# Potent neutralization by antibodies targeting the MPXV A28 protein

Ron Yefet [1,11], Leandro Battini[2,11], Mathieu Hubert[3], Katya Rakayev[1], Florence Guivel-Benhassine[3], Noam Rattner[4,5], Francoise Porrot [3], Lilach Abramovitz[1], Gilad Ostashinsky[1], Noam Ben-Shalom[1], Jeanne Postal[3], Ksenia Polonsky [6], Maya Ralph-Altman[1], Sireen Sweed[1], Tal Korner[1], Nadav Friedel[7], David Hagin[8], Eli Sprecher [7,9], Zvi Fishelson[10], Oren Kobiler [1], Lihi Adler-Abramovich [4,5], Olivier Schwartz [3,12] ✉, Pablo Guardado-Calvo [2,12] ✉ & Natalia T. Freund [1,12] ✉

Monkeypox virus (MPXV) is the most pathogenic Poxvirus in circulation, yet key viral antigens remain immunologically unexplored. We isolate and characterize a panel of monoclonal antibodies (mAbs) targeting MPXV A28 (OPG153), an important membranal protein present on mature MPXV virions. From male convalescent individuals, we isolate anti-A28 mAbs alongside additional mAbs targeting the A35 and H3 proteins. Anti-A28 mAbs potently neutralize MPXV and Vaccinia virus (VACV) through complement-dependent mechanisms involving C1q and C3 deposition. High-resolution crystal structures of two anti-A28 mAbs, 10M2146 and 8M2110, in complex with VACV A26 reveal two distinct and highly conserved proximal epitopes within the N-terminal domain. Passive transfer of 8M2110 modestly attenuates disease in infected female mice. Moreover, immunization with A28 elicits antigen-specific B cells and robust neutralizing antibody responses and provides protection against lethal VACV challenge. These findings identify MPXV A28 as a promising central target for the induction of neutralizing antibodies and antiviral interventions.

Monkeypox virus (MPXV) is a member of the Poxvirus family, an infamous viral lineage which has been a major source of human morbidity throughout history consequently termed Mpox[1,2]. Ever since the successful eradication of Variola virus (the causative agent of Smallpox) in 1980, MPXV has emerged as the leading Poxvirus threat in circulation. Furthermore, cessation of mass vaccination campaigns has left most of the population increasingly vulnerable to Poxvirus infection[3,4]. Consequently, Mpox cases have steadily increased in endemic African countries, and eventually spread to non-endemic countries[1,4–6]. In 2022, the emergence of a new MPXV sub-clade, IIb, led

[1]Department of Clinical Microbiology and Immunology, Gray Faculty of Medical and Health Sciences, Tel Aviv University, Tel Aviv, Israel. [2]G5 Structural Biology of Infectious Diseases, Institut Pasteur, Université Paris Cité, Paris, France. [3]Virus & Immunity Unit, Institut Pasteur, Université Paris Cité, Paris, France. [4]Department of Oral Biology, The Goldschleger School of Dental Medicine, Gray Faculty of Medical and Health Sciences, Tel Aviv University, Tel Aviv, Israel. [5]Jan Koum Center for Nanoscience and Nanotechnology, Tel-Aviv University, Tel-Aviv, Israel. [6]The Shmunis School of Biomedicine and Cancer Research, George S. Wise Faculty of Life Sciences, Tel Aviv University, Tel Aviv, Israel. [7]Division of Dermatology, Tel-Aviv Sourasky Medical Center, Tel Aviv, Israel. [8]Department of Immunology, Tel Aviv Sourasky Medical Center, Tel Aviv, Israel. [9]Department of Human Molecular Genetics and Biochemistry, Gray Faculty of Medical and Health Sciences, Tel-Aviv University, Tel Aviv, Israel. [10]Department of Cell and Developmental Biology, Gray Faculty of Medical and Health Sciences, Tel Aviv University, Tel Aviv, Israel. [11]These authors contributed equally: Ron Yefet, Leandro Battini. [12]These authors jointly supervised this work: Olivier Schwartz, Pablo Guardado-Calvo, Natalia T. Freund ✉e-mail: olivier.schwartz@pasteur.fr; pablo.guardado-calvo@pasteur.fr; nfreund@tauex.tau.ac.il

to swift dissemination of the virus to more than 100 non-endemic countries while infecting over 100,000 individuals[7]. To mitigate Mpox spread, an attenuated Vaccinia virus (VACV) based vaccine, Modified Vaccinia Ankara (MVA), was deployed to immunize at risk populations[8]. Although MVA confers protection from Mpox, new studies indicate a rapid decline of the elicited antibody response in the ensuing months, likely attributed to the attenuated nature and antigenic distance of MVA compared to MPXV[9–14]. This, along with the rapid emergence in 2024 of the more lethal MPXV sub-clade, Ib[15], underscores the urgent need to deepen our understanding of MPXV immune targets—a critical step toward developing more effective prevention and treatment strategies.

MPXV infection and vaccination elicits antibodies directed primarily against the surface proteins of its two infectious forms, Mature virions (MV) or Enveloped virions (EV), which are responsible for inter-host and intra-host transmission, respectively[16–23]. Out of the 30 different surface antigens expressed on the virus, eight were so far described as targets for neutralizing monoclonal antibodies (mAbs). These include the MV attachment and fusion proteins A29 (OPG 154), E8 (OPG 120), H3 (OPG 108), M1 (OPG 95), A16 (OPG 143), G9 (OPG 94) and the EV proteins A35 (OPG 161) and B6 (OPG 190—MPXV nomenclature)[22,24–30]. Some of these mAbs confer protection in animal models, most effectively when administered in a combination targeting both forms of the virus[22,31,32]. Moreover, the activity of human Poxvirus targeting mAbs is often complement-dependent[22,26]. The sheer abundance of viral targets recognized by antibodies elicited during infection, coupled with their largely unexplored mechanisms of viral neutralization, highlights the importance of identifying additional neutralizing antibodies targeting new viral sites and further exploring their mechanisms of action. Recent immunization studies in mice have indicated that additional MPXV surface proteins could serve as targets for neutralizing antibodies[33]. Among these, A28—a major structural component of the MV and a target of antibodies elicited by infection or vaccination—emerges as a promising candidate[23,34]. Acting as a key membranal protein, A28 facilitates viral attachment through laminin binding and controls endosomal entry by regulating the viral fusion machinery[19,35,36]. While A28 represents a promising target for neutralizing antibodies, mAbs against this protein were not yet reported.

In this study, we characterize human monoclonal neutralizing antibodies targeting the MPXV A28 antigen. By analyzing B cells from Mpox convalescent donors, we generated a panel of mAbs against A28, as well as the previously described MPXV antigens, A35 and H3. Among these, mAbs targeting the A28 antigen demonstrated the highest neutralization, which was complement-dependent. Using X-ray crystallography, we obtained high-resolution structures of the two most effective anti-A28 mAbs, 10M2146 and 8M2110, in complex with the VACV A28 homolog, A26, showing that they bind two proximal, highly conserved, non-overlapping epitopes. The mode of action of both A28-targeting mAbs was elucidated using functional studies and immunogold transmission electron microscopy (TEM). Despite its potent in vitro neutralizing activity, passive administration of mAb 8M2110 in VACV-infected mice yielded only modest, non-significant clinical benefit in vivo. Nonetheless, A28 mice immunization elicited robust antigen-specific B cell responses and high-titer neutralizing antibodies, exhibiting both complement-dependent and -independent activities, and conferred complete protection against a lethal VACV challenge. Overall, our study presents A28 as a potential vaccine modality for elicitation of pan-Poxviruses neutralizing antibodies and provides functional and mechanistic insights about how human antibodies recognize and neutralize MPXV.

## Results

### Generation of mAbs from human Mpox convalescent donors
Mpox infection elicits antibodies targeting both the MV and EV forms[16–18]. We previously reported that Mpox convalescent donors

infected during the outbreak of May-June 2022 developed strong antibody and B cell responses against the A35 and H3 antigens[18]. Here, we isolated monoclonal antibodies (mAbs) from the same cohort while focusing on MPXV antigen A28 (OPG153), in addition to A35 (OPG 161) and H3 (OPG 108). To assess the serological response to A28, we recombinantly expressed residues 2-361 in a mammalian protein expression system (Fig. 1a,b) and tested serum reactivity of our cohort. A28 was strongly bound by 9 out of 11 convalescent sera (Supplemental Fig. 1). Next, to isolate MPXV-specific B cells, we employed flow cytometry-based staining on selected convalescent donor samples collected 1-2-months post infection (indicated as V1, Supplemental Table 1) or 9-10-months post infection (indicated as V2, Supplemental Table 1). The peripheral blood mononuclear cells were stained with anti-CD19, anti-IgG and duo-labeled A28, A35 or H3[18] (Fig. 1c). The identified cells were collected by single cell sorting, the mRNA of each B cell was extracted, and the membrane immunoglobulin heavy and light chains ($Ig_H$ and $Ig_L$, respectively) were amplified by PCR and Sanger-sequenced, as previously reported[37–40]. High-quality paired $Ig_H$/$Ig_L$ sequences were cloned and expressed as IgG1 mAbs. Four mAbs demonstrated binding to A28 (the mAbs 10M2146 and 10M2154, were clonal relatives), nine mAbs to A35 (the mAbs, 4M1130 and 4M1224, and the pair of mAbs, 4M1133 and 4M1166, were clonally related), and five mAbs to H3 (Supplemental Data 1, Fig. 1d,e).

We next tested whether the MPXV mAbs could also recognize their corresponding Vaccinia virus (VACV) homologs: A26 (A28, OPG 153), A33 (A35, OPG 161) and H3 (H3, OPG 108) all of which share a high sequence similarity (Fig. 1a,b)[41,42]. Most antibodies showed cross reactivity: three out of four A28 mAbs, six out of nine A35 mAbs and four out of five H3 mAbs, bound to both MPXV and VACV antigens. The remaining mAbs displayed preferential binding to either the MPXV or VACV orthologs, indicating recognition of species-specific domains (Fig. 1e-g). Competition ELISA pointed out that three of the four A28 mAbs (8M2110, 10M2146 and 10M2154) compete with one another, and therefore likely target proximal sites. Similarly, the nine A35 mAbs clustered into two main epitope groups, primarily based on their differential binding to the MPXV and VACV homologs. Two of the five H3 mAbs competed with each other, while the rest did not (Fig. 1h-j). This suggests that Mpox infection elicits antibodies directed against convergent immunodominant sites on A28, A35 and H3 antigens. These sites are partially conserved between VACV and MPXV antigen homologs, highlighting both shared and species-specific antigenic determinants.

The mAbs exhibited moderate levels of somatic hypermutations (SHMs, Supplemental Data 1). To explore whether affinity maturation plays a role in the antibody response to A28, A35, and H3, we selected nine mAbs representing the major antigenic clusters (based on competition ELISA) and reverted them to their predicted germline versions (indicated as 'Germlines 1-6', Fig. 1k-m). Predicted germlines 1-6 were generated based on $V_HD_HJ_H$ and $V_LJ_L$ IMGT alignment, while keeping the CDRH3 intact as was previously reported[43–45]. When clonally related sequences exhibited divergence in their CDRH3, the $D_H$ segment was used to assign the most likely germline origin. Our data indicates that mostly the binding of A28-targeting mAbs does not relay on SHM. As such, Germline 1, in which the SHMs of the clonally related anti-A28 mAbs 10M2146 (10 and 5 amino acid substitutions in $Ig_H$ and $Ig_L$, respectively) and 10M2154 (3 amino acid substitutions in $Ig_H$, none in $Ig_L$) were reverted, demonstrated similar binding to either A28 or the VACV homolog. Germline 2—the germline-reverted 8M2110 (with 7 and 3 amino acid substitutions in $Ig_H$ and $Ig_L$, respectively)—lost its binding to A28 yet retained its binding to the VACV homolog A26 (Fig. 1k-m). These findings suggest that affinity maturation contributed mostly to the cross reactivity with other Poxviruses, while the ability to target the particular antigenic site is likely to be germline-encoded. A mixed pattern was observed with mAbs targeting the A35 antigen. Germline 3, the unmutated precursor of mAbs 4M1130 and 4M1224 (each

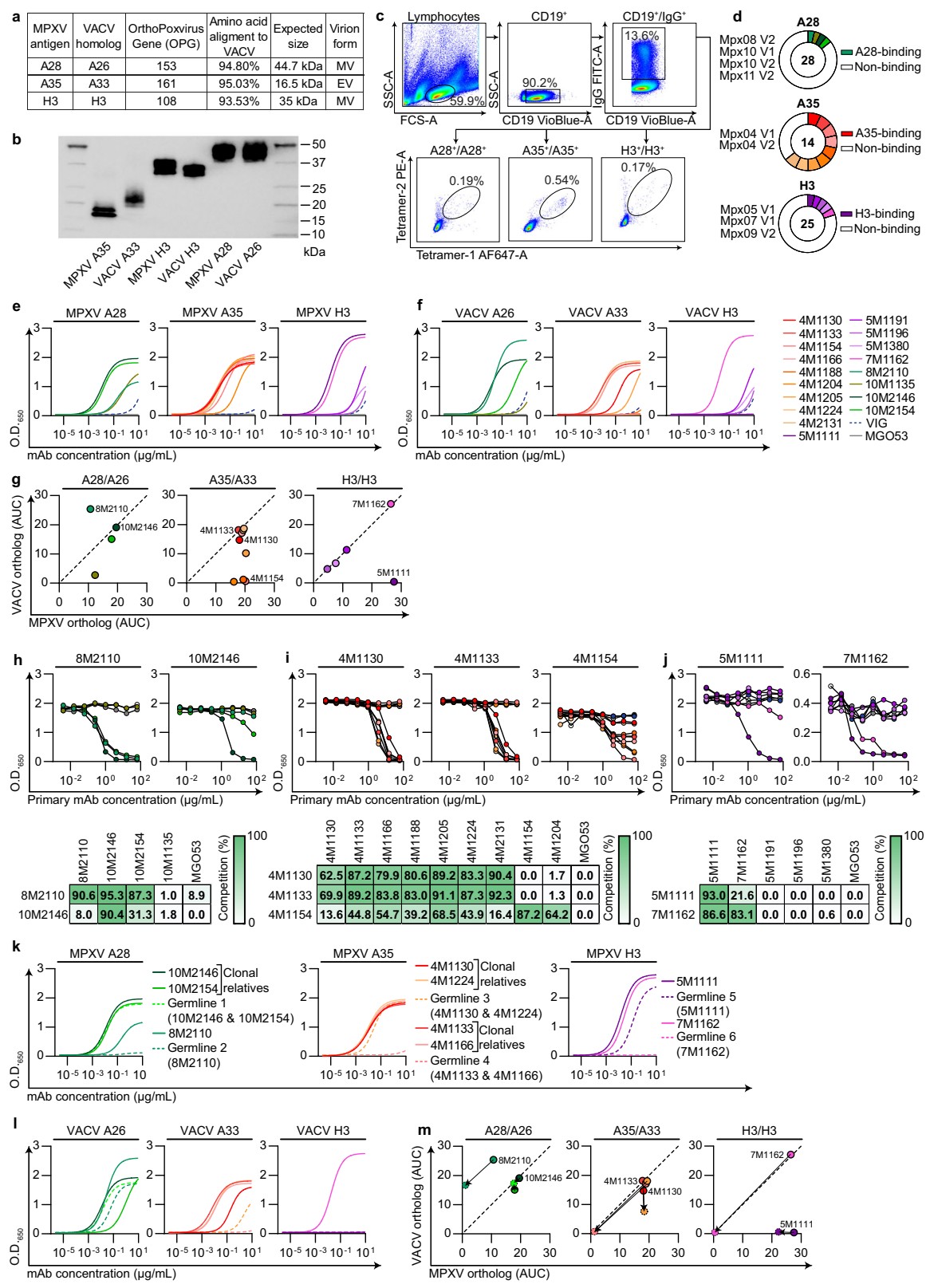

harboring only 2 amino acid substitutions in the heavy chain), showed only a slight reduction in binding compared to their mature counterparts. In contrast, Germline 4—corresponding to mAbs 4M1133 (with 6 and 1 amino acid substitutions in the heavy and light chains, respectively) and 4M1166 (7 substitutions in the heavy chain)—completely lost binding capacity towards A35 (Fig. 1k-m). A similar trend was noted

for anti-H3 mAbs: Germline 5, derived from mAb 5M1111 (5 and 3 substitutions in heavy and light chains), retained binding, while Germline 6 of mAb 7M1162 (also with 5 and 3 substitutions) showed a complete loss of reactivity. These findings suggest that while some MPXV-specific antibodies are germline-encoded, others require SHM for effective binding.

**Fig. 1 | Generation and characterization of anti-A28, A35, and H3 MPXV mAbs.** **a** MPXV antigens and corresponding VACV homologs used in this study. **b** Western blot of MPXV antigens and VACV homologs under reducing conditions, detected with anti-Avi-tag HRP. **c** Representative gating strategy for sorting antigen-specific B cells from PBMCs of Mpox convalescent donors Mpx10 V2, Mpx04 V1, Mpx05 V1 using antigens A28, A35, H3, respectively. **d** Pie charts showing total IgG1 mAbs generated against each antigen indicated by the number in center; white sections indicate non-binding mAbs, colored slices indicate mAbs that exhibit binding by ELISA; donor origin indicated. **e–f** ELISA binding curves for anti-MPXV mAbs against MPXV antigens (**e**) or VACV homologs (**f**) across 12 consecutive 4-fold dilutions, starting from 10 µg/mL. **g** Differential two-dimensional representation of the AUC values from (**e–f**); dashed line represents line of identity. **h–j** Competition ELISA for anti-A28 (**h**), anti-A35 (**i**), and anti-H3 (**j**) mAbs; binding to antigen from 8 consecutive 4-fold dilutions in the presence of 0.2 µg/mL biotinylated competitor as

indicated; percent competition shown in accompanying table. For (**h**): 8M2110 or 10M2146; for (**i**): 4M1130, 4M1133, or 4M1154; for (**j**): 5M1111 or 7M1162. **k–l** ELISA binding curves for germline-reverted anti-MPXV mAbs against MPXV antigens (**k**) or VACV homologs (**l**) across 12 consecutive 4-fold dilutions, starting from 10 µg/mL. **m** Differential two-dimensional representation of the AUC values from (**k–l**); dashed line represents line of identity; arrows indicate transitions from unmutated to germline-reverted anti-MPXV mAbs in dotted lines. Anti-A35, anti-H3, and anti-A28 mAbs are shown in red, purple, and green, respectively; VIG dashed blue; MGO53 isotype control in gray; germline-reverted mAbs in (**k–l**) are dashed in corresponding colors. Binding curves were determined by fitting values using sigmoidal four-parameter 4PL nonlinear regression (X is concentration) for panels (**e, f, k, l**). Anti-MPXV mAbs are shown in panels (**e–m**) in left-to-right order as anti-A28, anti-A35, and anti-H3. Data are representative of two independent experiments with similar results (**b, e-m**). Source data are provided as a Source Data file.

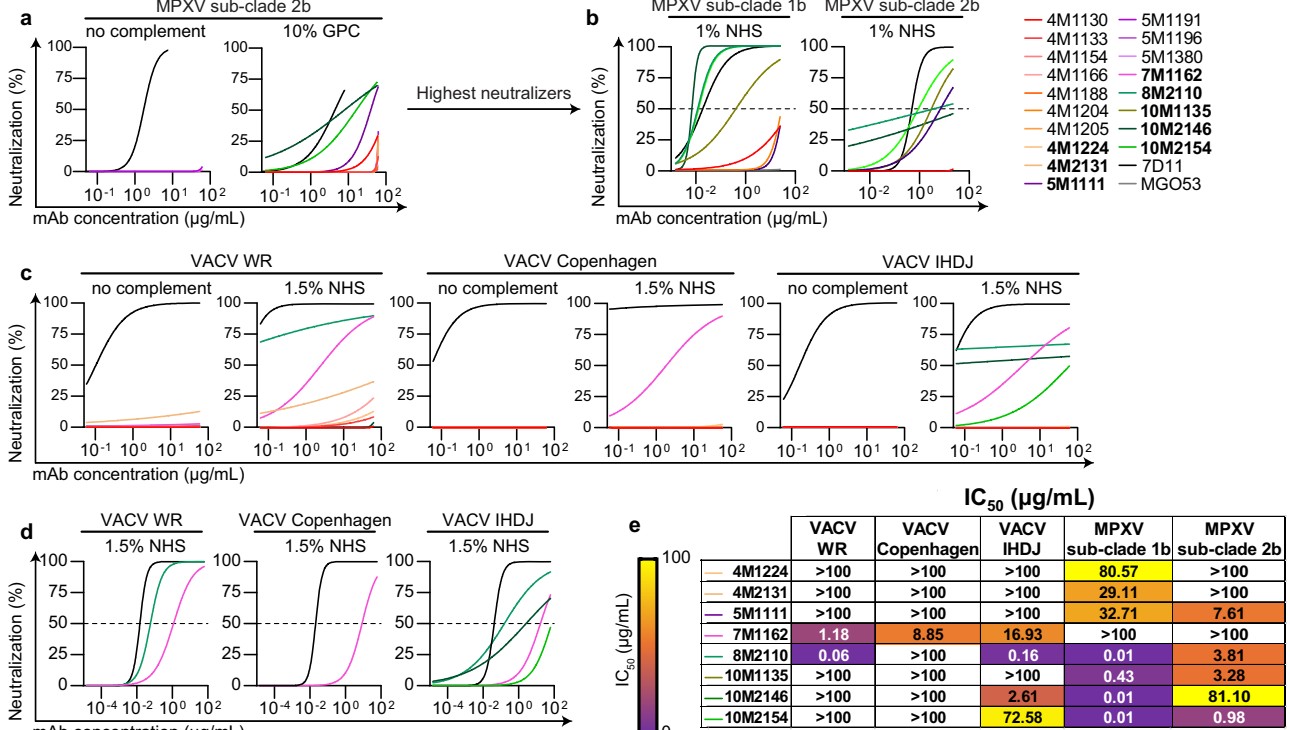

**Fig. 2 | Anti-MPXV mAbs neutralization is complement dependent. a** Anti-MPXV mAbs (excluding anti-A28 mAbs 8M2110 and 10M1135) neutralization of MPXV sub-clade IIb in the absence and presence of 10% GPC (20% during pre-incubation). Neutralization curves of 6 consecutive 4-fold dilutions, starting from 62.5 µg/mL. **b** Potent anti-MPXV mAbs neutralization of MPXV sub-clades Ib and IIb in the presence of 1% NHS (2% during pre-incubation). Neutralization curves of 8 consecutive 4-fold dilutions, starting from 25 µg/mL. Dashed line represents a threshold of 50% neutralization. **c** Anti-MPXV mAbs neutralization of VACV strains in the absence and presence of complement. Neutralization curves of 6 consecutive 4-fold dilutions, starting from 62.5 µg/mL. Panels represent neutralization of VACV WR, VACV Copenhagen, and VACV IHDJ strains, from left to right. **d** Potent anti-MPXV mAbs neutralization of VACV strains in the presence of 1.5% NHS (3% during

pre-incubation). Neutralization curves of 12 consecutive 4-fold dilutions, starting from 62.5 µg/mL. Dashed line represents a threshold of 50% neutralization. **e** Neutralization IC$_{50}$ values from (**b**) and (**d**); scale bar depicting IC$_{50}$ values color code is shown on the left. Anti-A35, anti-H3, and anti-A28 mAbs are shown in red, purple, and green, respectively; anti-M1 (7D11) as a positive control in black; MGO53 isotype control in gray. Names of neutralizing mAbs are in bold. Neutralization curves and IC$_{50}$ values were determined by fitting values using Agonist vs. normalized response (Variable slopes) nonlinear regression. Percentages of complement sources indicated represent the final concentrations in the infected wells. Data are representative of two independent experiments with similar results. Source data are provided as a Source Data file.

## Antibodies targeting MPXV A28 exhibit potent complement-dependent neutralization

We next evaluated the neutralizing capacity of our mAbs against MPXV (sub-clades Ib and IIb), as well as against three VACV strains, WR, Copenhagen, and IHDJ using virus preparations enriched for MV. Several mAbs demonstrated neutralizing activity against MPXV or VACV, which in all cases was complement-dependent, except for the anti-A35 mAb 4M2131 that demonstrated weak but detectable neutralization against VACV WR without complement (Fig. 2c).

Complement-dependent neutralization is consistent with previous reports of human antibodies elicited following infection[18,22,26,46]. Among the tested mAbs, those targeting A28 exhibited the most potent neutralization, though their efficacy varied between MPXV sub-clades and VACV strains. For the anti-A28 mAbs: the clonal pair 10M2146 and 10M2154 neutralized MPXV sub-clade Ib with IC$_{50}$ values of 0.01 µg/mL for both mAbs, and sub-clade IIb with IC$_{50}$ values of 81.1 µg/mL and 0.98 µg/mL, respectively. MAb 8M2110 neutralized MPXV sub-clade Ib with an IC$_{50}$ of 0.01 µg/mL and sub-

clade IIb with an $IC_{50}$ of 3.81 µg/mL. Meanwhile, the mAb 10M1135 neutralized MPXV sub-clade Ib with an $IC_{50}$ of 0.43 µg/mL and sub-clade IIb with an $IC_{50}$ of 3.28 µg/mL (Fig. 2a,b,e). The mAbs also neutralized VACV: 10M2146 and 10M2154 neutralized the IHDJ strain with $IC_{50}$ values of 2.61 µg/mL and 72.58 µg/mL, respectively; 8M2110 neutralized WR and IHDJ with $IC_{50}$ values of 0.06 µg/mL and 0.16 µg/mL, respectively; whereas 10M1135 showed no detectable neutralization against any VACV strain (Fig. 2c-e). VACV Copenhagen was resistant to all tested anti-A28 mAbs, likely a result of the absence of the A26 antigen on the viral surface due to a truncation of its C-terminal region (Supplemental Fig. 3)[47–49]. The ability of anti-A28 mAbs to neutralize VACV WR and IHDJ, along with multiple MPXV sub-clades, underscores their potential as broadly protective therapeutics against Poxviruses.

Among the anti-H3 mAbs, 7M1162 was the only one to neutralize all three tested VACV strains, with $IC_{50}$ values of 1.18 µg/mL, 8.85 µg/mL, and 16.93 µg/mL against WR, Copenhagen, and IHDJ, respectively, but it did not neutralize either MPXV sub-clade. In contrast, the MPXV-specific mAb 5M1111 (anti-H3) neutralized MPXV with $IC_{50}$ values of 32.71 µg/mL and 7.61 µg/mL for sub-clades Ib and IIb, respectively, but had no activity against VACV (Fig. 2). As might be expected, in this assay, mAbs targeting A35 exhibited minimal detectable neutralization activity, with measurable $IC_{50}$ values only for mAbs 4M1224 and 4M2131 against MPXV sub-clade Ib (80.57 µg/mL and 29.11 µg/mL, respectively- Fig. 2). This is likely due to the assay's primary focus on MV as opposed to EV.

### MAb combinations targeting different surface proteins enhance MPXV neutralization
Simultaneous targeting of multiple antigens on the surface of viruses is advantageous[50], and was previously shown to enhance Poxvirus neutralization in vitro and in animal models[22,31,32]. To evaluate the efficacy of different mAb combinations, we tested our 18 mAbs in pairs. For MPXV, as well VACV strain IHDJ, combinations targeting 2 different antigens exhibited significantly stronger neutralization compared to combinations targeting a single antigen (Fig. 3a-d). This was not the case for VACV strains WR and Copenhagen. When the two most potent mAbs targeting different antigens were tested alone and in combination across 12 serial dilutions, nearly all combinations showed a significant reduction in $IC_{50}$ and $IC_{80}$ values compared to single mAbs (Fig. 3e and Supplemental Fig. 2a-c). Notably, a synergistic effect was observed between A28- and H3-targeting mAbs or the anti-A35 mAbs, even though the latter exhibited minimal neutralization on their own.

### 8M2110 and 10M2146 mAbs bind two proximal epitopes of the N-terminal domain on A28
To further characterize anti-A28 neutralizing mAbs, we performed structural analyses focusing on 8M2110 and 10M2146, each isolated from a different donor. For these studies, we utilized the VACV homolog of A28, A26. Structurally, A26 comprises two domains: the "head domain" (residues 1–320), which contains both the acid-sensing region (residues 1–75) and the fusion suppressor region (residues 76–320), both implicated in the pH-dependent entry of the virion; and the "tail domain" (residues 320–500), which includes the A27-binding region that anchors A26 to the viral membrane through its interaction with VACV A27 (homologous to MPXV A29)[51]. The structure of the head domain has been previously reported and subdivided into a N-terminal domain (NTD; residues 1–228) and a C-terminal domain (CTD; residues 229–364) (Supplemental Fig. 3)[35]. The NTD consists of twelve α-helices (α1–α12) and contains two histidine residues (H48 and H53), which are essential for viral infectivity. The CTD adopts a mixed α/β fold composed of six α-helices (α13–α18) and six β-strands (β1–β6)[35]. For structural determination, both mAbs were expressed as F(ab) fragments and complexed with the residues 1-397 of A26, encompassing the full fusion-regulatory region.

We obtained the crystal structures of 8M2110:A26 and 10M2146:A26 at 2.9 Å and 1.9 Å resolution, respectively (Fig. 4, Supplemental Table 2 and Supplemental Figs. 4 and 5). In both complexes, A26 resembles its unbound conformation (rmsd = 0.2 Å), with no significant antibody-induced conformational changes. The paratope of 10M2146 is comprised mainly of the complementarity-determining regions (CDRs) H2, H3, L2, and L3, and the N-terminus of the light chain, and spans a buried surface area of 971 Å² on the NTD, mainly involving helixes α7 and α8, with some minor contacts on α4 and α11 (Fig. 4a,b). Sequence analysis (Supplemental Fig. 3) indicates that two epitope residues differ in MPXV (F153 to L153, E156 to D156), but structural data suggest these substitutions are compatible with antibody binding. E156, which interacts with K78 and Y122 in the heavy chain, can be replaced by D156, and F153 can be replaced by L153 without introducing steric clashes (Fig. 4b). The same can be said of the three somatic mutations of the paratope, K78R, N121Y and A125S (Supplemental Fig. 6a-b), which do not generate apparent clashes or electrostatic repulsions. The paratope of 8M2110 is formed mainly by CDRs H1, H2, and H3, with little to no contribution from the light chain. The epitope is spanning over 843 Å², mostly (90%) on the NTD, involving residues in helices α10-13 (Fig. 4c,d). Sequence analysis of the epitope revealed two residues that differ between VACV A26 and MPXV A28: A207 (V207 in MPXV) and F235 (L235 in MPXV) (Supplemental Fig. 3). Substituting A207 with V207 is expected to introduce some steric clashes with CDR-H1, which may explain why 8M2110 binds VACV A26 better than MPXV A28 (Figs. 1e-g, 4d and Supplemental Fig. 6). Indeed, the 8M2110 germline has a mutation close to CDR-H1, in position 50, that could induce a conformational change incompatible with the presence of V207, which would explain why the germline does not bind A28 (Supplemental Fig. 6). Mapping the footprints of both antibodies on A26 (Fig. 4e) reveals that they bind two distinct epitopes that are very close to one another, with some overlap on helix α11. Superimposition of the complexes demonstrates a steric clash between the variable domain of the 8M2110 light chain and the variable domain of the 10M2146 heavy chain (Fig. 4f), thus explaining the inability of 8M2110 to bind 10M2146 pre-bound A28 in ELISA (Fig. 1h).

The binding epitope of the anti-H3 mAb 7M1162 was predicted using 11 affinity selected peptides isolated from screening random phage display peptide libraries followed by computational prediction[52–55]. This approach generated two predicted clusters (A and B) (Supplemental Fig. 7a-c). Point mutation in position 167 within Cluster A $H3_{H167G}$, $H3_{H167K}$ exhibited reduced to complete loss of binding to 7M1162, thus supporting the critical role of this residue in the antibody binding site (Supplemental Fig. 7d). Notably, the H167K or H167G substitution did not affect binding of 5M1111.

### Anti-A28 mAbs block viral infection by recruiting C1q and C3
Consistent with previous reports on anti-Poxvirus antibodies[18,22,26,46,56,57], most of the mAbs isolated in this study, required complement to effectively neutralize either VACV or MPXV. Moreover, heat inactivation at 56 °C of the normal human serum (NHS) used as a source of complement in our study abolished neutralization (Supplemental Fig. 8a). We therefore sought to decipher the mechanism of complement-dependent neutralization. For these assays we used VACV IHDJ, and the neutralizing mAbs 7M1162 (anti-H3) and 8M2110, 10M2146 and 10M2154 (anti-A28). The strongest neutralizing effect was observed when antibodies were pre-incubated with the virus and 1.5% NHS for two h before infection. Shortening this pre-incubation time drastically reduced antibody neutralization, resulting in 40% - 100% reduction when mAbs and NHS were added to U2OS cells at the time of infection (Fig. 5a). These results suggest that the antibodies primarily act on the virus itself, rather than on infected cells, and that their neutralization potency builds up over time. Supporting this, despite binding to infected cells, at 1.5% NHS, anti-MPXV mAbs did not trigger complement-

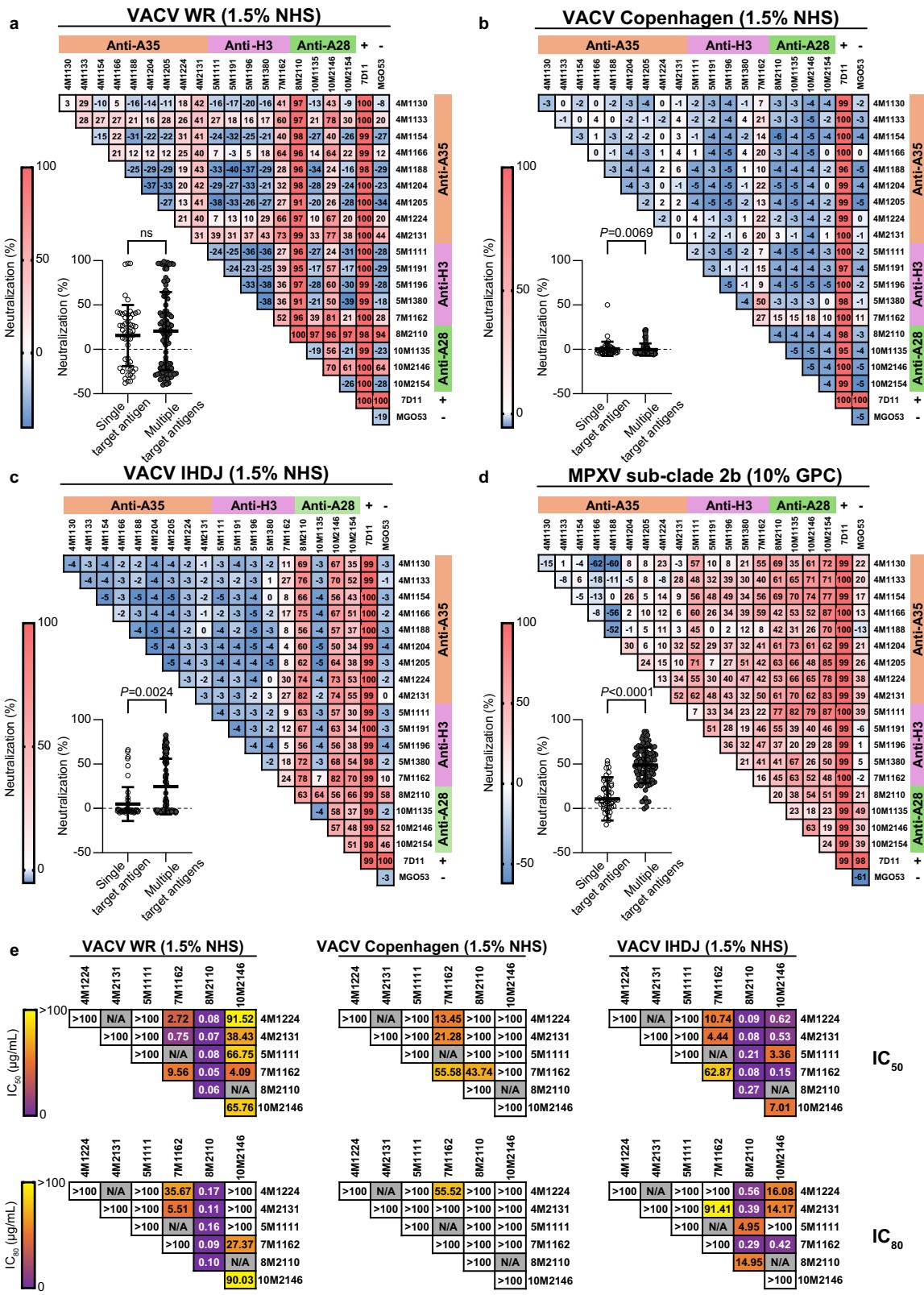

dependent cytotoxicity (Supplemental Fig. 8b,c and 11b). We next investigated whether the mAbs promote complement-dependent virolysis, a phenomenon previously reported for the EV form of VACV[58]. However, inhibition of C5 using C5-IN-1 (Compound 7) had no impact on neutralizing activity of our mAbs, suggesting that formation of membrane attack complex is unlikely to be the primary mechanism of antibody-mediated neutralization (Fig. 5b). In

contrast, blocking C3 with the selective inhibitor AMY-101 (CP40) significantly reduced neutralizing activity (Fig. 5c).

The mAbs effectively bound the viral surface and promoted C1q deposition (Fig. 5e and Supplemental Fig. 8d), however, exogenous C1q alone was insufficient for antibody-mediated neutralizing activity, suggesting that both C1q and C3 are essential for viral neutralization (Supplemental Fig. 8e and 10a). Serum depletion of either C1q or C3

**Fig. 3 | Combinations of anti-MPXV mAbs enhance poxvirus neutralization. a** Heatmap showing neutralization of VACV WR using combinations of anti-MPXV mAbs in the presence of 1.5% NHS (3% during pre-incubation) at a total mAb concentration of 62.5 µg/mL. Neutralization values were normalized to baseline infection (no mAb). The neutralization scale bar is shown on the left. Lower left panel: analysis of neutralization values for all independent anti-MPXV mAb combinations (excluding 7D11 and MGO53) targeting single or multiple MPXV antigens. Data are presented as mean values ± standard deviation for each group ($n = 52$ for single antigen targeting by two-mAb combinations; $n = 52$ for multiple antigen targeting by two-mAb combinations). **b** Same as (**a**) but for VACV Copenhagen. **c** Same as (**a**) but for VACV IHDJ. **d** Same as (**a**) but for MPXV in the presence of 10% GPC (20% during pre-incubation). **e** Heatmap summarizing $IC_{50}$ and $IC_{80}$ values of selected two-mAb combinations targeting two distinct antigens, as shown in Supplemental Fig. 2. Panels represent VACV WR, VACV Copenhagen, and VACV IHDJ strains, from left to right. Scale bar is shown on the left. For panels (**a**–**d**), mAb combinations targeting a single MPXV antigen are shown as clear dots, and combinations targeting multiple antigens are shown as dark gray dots. Statistical analysis was performed using two-tailed Mann–Whitney test. Standard deviations of the mean are shown for all values corresponding to single-antigen ($n = 52$ independent two-mAb combinations) or multiple-antigen ($n = 101$ independent two-mAb combinations) targeting. Negative neutralization values indicate combinations with higher infection levels than the baseline infection control. The percentages of complement sources indicated represent the final concentrations in the infected wells. ns = not significant. Source data are provided as a Source Data file.

impaired neutralization efficacy (Fig. 5d), further strengthening our conclusion that mAb-mediated neutralization is facilitated by deposition of both C3 and C1q. TEM imaging confirmed the presence of IgG, C1q, and C3 on the viral surface (Fig. 5f-h), supporting a previously described mechanism named 'full occupancy model'[56], in which mAbs and C3 act as 'molecular handcuffs' to immobilize the virus and prevent cellular entry.

### Active vaccination with A28 protein mediates protection from lethal viral challenge in vivo

Prophylactic administration of 8M2110 (anti-A28) to female BALB/c mice (Supplemental Fig. 9a) did not confer protection from a lethal VACV challenge. Nevertheless, treated mice displayed a modest, although not statistically significant, delay in the onset of severe disease (weight $P = 0.056$, survival $P = 0.1013$), along with a slight reduction in viral load on day 5 post-infection compared to mice treated with isotype control antibody $P = 0.0883$ (Supplemental Fig. 9b-d). This trend was reproducible across three independent experiments. The gap between the potent in vitro neutralization capacity of 8M2110 and its limited in vivo efficacy is likely the result of the dominant EV form in intra-host viral dissemination, where the A28 antigen is not exposed.

To assess whether vaccination with A28 elicits neutralizing antibodies capable of conferring protection, we immunized female mice three times (both BALB/c and C57BL/6, see Methods) with either PBS, MPXV A28, or VACV A26 antigens formulated with QS-21 (saponin) or Alum, respectively (Fig. 6a and Supplemental Fig. 10b). Immunization with MPXV A28 conferred protection against a lethal VACV challenge (Fig. 6b-c and Supplemental Fig. 10a). To assess whether this protection is mediated by antibodies we analyzed mouse sera and B cells. As expected, antigen-specific antibody titers increased following each immunization (Supplemental Fig. 10c). Furthermore, sera from vaccinated mice effectively neutralized both VACV WR and IHDJ strains (as anticipated, VACV Copenhagen was not neutralized, Supplemental Fig. 10d). Surprisingly, heat-inactivated sera retained neutralizing activity almost completely even in the absence of complement, suggesting that antibodies elicited following A28 immunization block viral attachment or entry through additional, complement-independent mechanisms (Fig. 6d). Further characterization of the immune cell response of immunized mice revealed elevated levels of germinal center (GC) B cells and an increased frequency of A28-binding IgG$^+$ B cells (Supplemental Fig. 10e,f), while no significant differences were observed in the total B cell (B220+), CD4+, or CD8+T cell populations between the immunized groups (Supplemental Fig. 10e) Notably, immunization with VACV A26 provided less protection than MPXV A28. Both BALB/c and C57BL/6 mice showed lower neutralizing antibody titers following A26 vaccination. Among the four A26-vaccinated mice that did not survive, neutralizing antibody levels were particularly low within their group. Together, these findings demonstrate that immunization with MPXV A28 induces a robust humoral immune response, marked by the production of neutralizing antibodies that

act through both complement-dependent and -independent mechanisms, ultimately conferring effective protection against lethal Poxvirus challenge.

## Discussion

Mpox outbreaks have become increasingly frequent in recent years, highlighting the need for novel targets for antibody-based interventions. In this study, we identify A28 protein as a dominant target of human neutralizing antibodies. Although A28 is not essential for infection, it intrinsically regulates the transition from EV production, associated with intra-host spread, to MV production, which facilitates inter-host transmission[59]. In addition, A28 promotes viral dissemination by mediating attachment through laminin binding and by suppressing premature activation of the entry fusion complex under low-pH conditions. Together, these multifaceted functions position A28 as a key contributor to viral spread and inter-individual transmission. The strong antibody response elicited following Mpox infection or vaccination further underscores its potential as a promising antigenic target for antibody-based interventions. The mAbs isolated from convalescent donors exhibited high-affinity binding to MPXV A28 and its VACV homolog A26 and demonstrated potent neutralization of MPXV and multiple VACV strains. Structural analysis revealed that the neutralizing mAbs bind to distinct, proximal epitopes on the NTD of A28, providing insight into conserved immunogenic regions, and supporting A28 classification as a promising candidate for broad Poxvirus vaccine development. The identification of A28 as a potent target for neutralizing antibodies against MPXV advances our understanding of immune responses to Poxviruses and informs future therapeutic and vaccine strategies.

Anti-viral neutralizing mAbs often neutralize by blocking viral entry into host cells[60]. However, unlike viruses such as HIV-1, SARS-CoV-2, and influenza, where antibodies alone can effectively prevent infection[37,61–63], Poxvirus neutralization frequently depends on complement factors. This complement-dependent mechanism, long recognized as a key feature of Poxvirus immunity[22,46,56,57], was consistently observed among the potent anti-A28 and anti-H3 mAbs characterized here. Our mechanistic data demonstrates that antibody-mediated viral neutralization is facilitated by the deposition of complement components C1q and C3 on the viral surface. These complement factors likely enhance antibody effectiveness by overcoming the structural complexity of the Poxvirus entry machinery. Given the large size of Poxviruses and multiple pathways for host cell entry, complement deposition likely serves as an amplifier, working with antibodies to ensure efficient blockade of infection. TEM analysis further provided compelling visual confirmation of this "molecular handcuff" model, complying with the full occupancy model, in which antibodies immobilize virions via complement engagement, thereby preventing infection[56,57,60]. Interestingly, while complement was essential for most mAbs, immunization experiments revealed that A28 can also elicit complement-independent neutralizing antibodies. This dichotomy suggests that A28 harbors multiple neutralizing epitopes with diverse functional properties, highlighting the potential for inducing robust

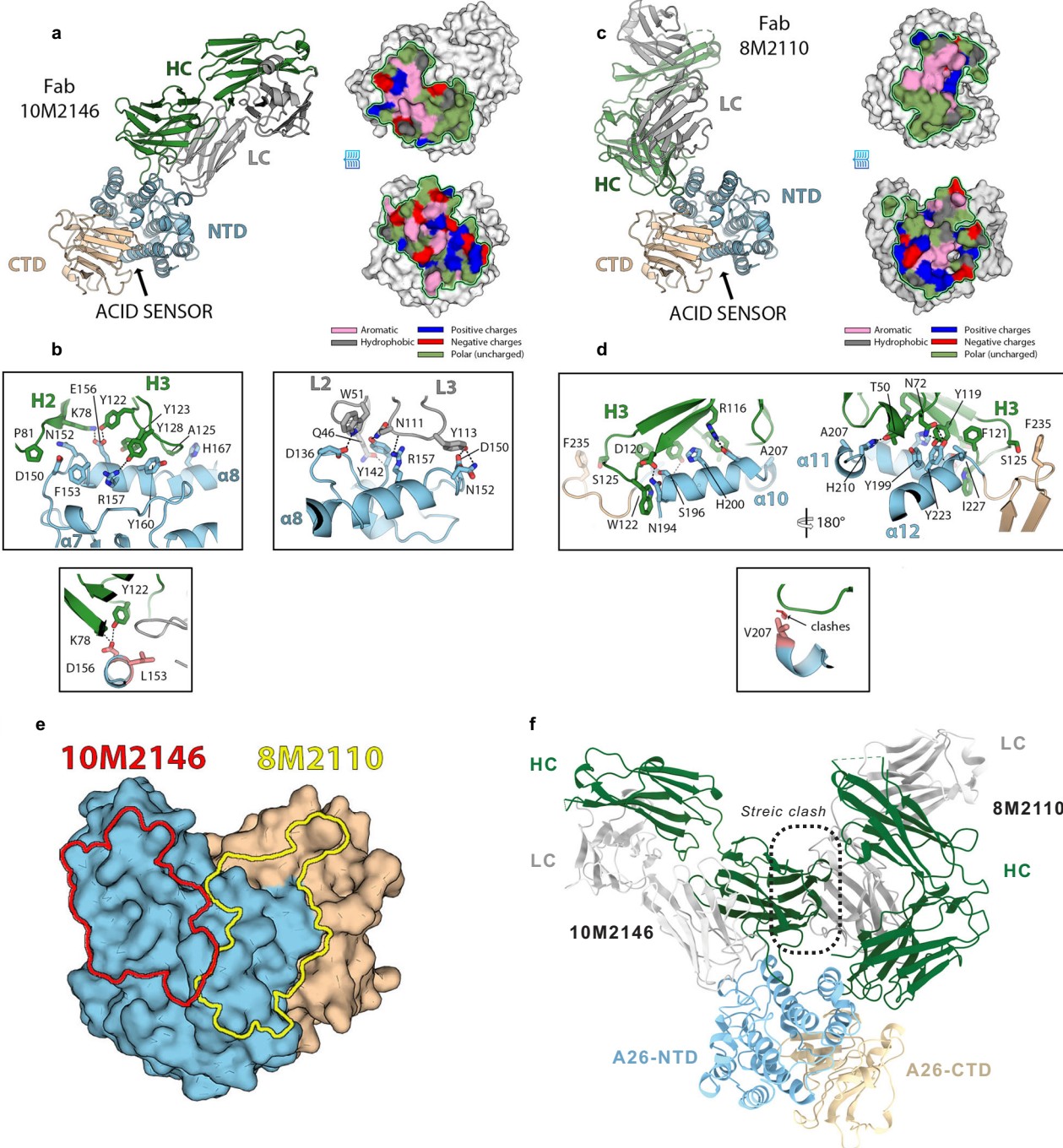

**Fig. 4 | Crystal structures reveal how 10M2146 and 8M2110 recognize distinct A26 epitopes. a** Structure of the VACV A26/Fab-10M2146 complex. Left panel: The VACV A26 NTD and CTD are shown in blue and tan, respectively. The heavy chain of the Fab is colored green, the light chain in gray. An arrow marks the approximate location of the pH-sensing histidines. Right panel: "open book" views of the paratope (top) and epitope (bottom) outlined with a green line and colored based on the chemical nature of the residues, as indicated. **b** Close-up views of the 10M2146 epitope with the side chains involved in the interaction showed in sticks and labelled. Upper panels: interactions involving CDR H2 and H3 (left) and interactions involving CDR L2 and L3 (right). Lower panel: display of the two epitope residues

that differ between MPXV and VACV, colored red and labelled, which have no impact on the interaction with the antibody. **c** Structure of the VACV A26/Fab-8M2110 complex colored as in (**a**). **d** Upper panel: Two close-up views of the 8M2110 epitope with the CDR H3 side chains involved in the interaction showed in sticks and labelled, as in (**b**). Lower panel: displays the epitope residue V207 present in MPXV, which clashes with the antibody main chain. **e** Outline of the footprints of 10M2146 (red) and 8M2110 (yellow) on VACV A26. **f** Superimposed structures of A26 bound simultaneously to 8M2110 and 10M2146. A steric clash occurs between the light chain of 8M2110 and the heavy chain of 10M2146 (indicated by the boxed region).

immunity through targeted antigen design. Another key finding is the synergy achieved by combining mAbs targeting A28 with anti-H3 or anti-A35 mAbs. Targeting different antigens enhanced neutralization breadth and potency, highlighting the value of multi-targeted therapeutic strategies.

Finally, our study demonstrates the feasibility of rational vaccine design based on A28, with vaccinated mice showing protection from a lethal VACV viral challenge. Vaccinations remain the most efficient method for preventing infections or complications from viral diseases. Historically, Poxviruses were the first pathogens to be

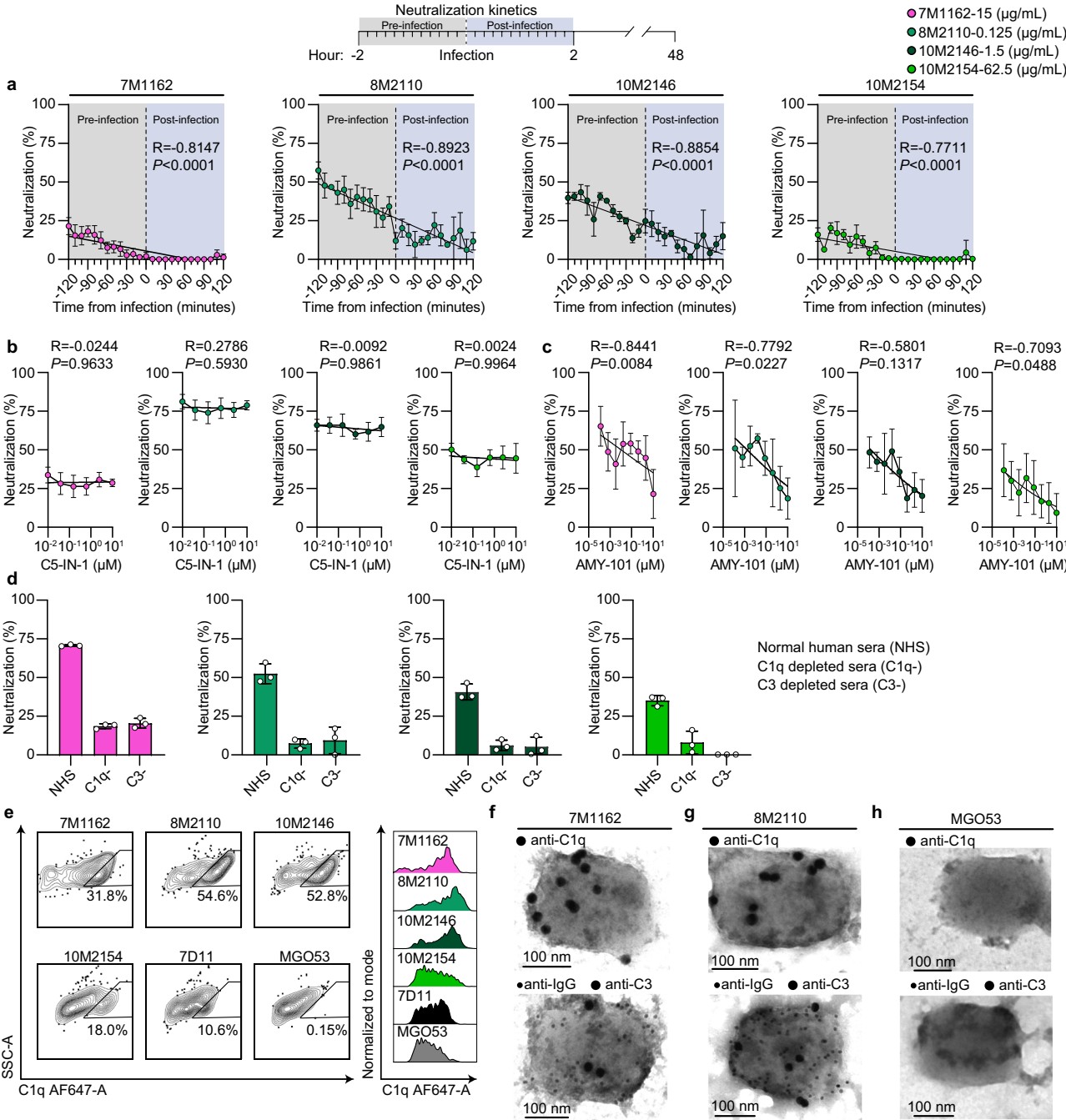

**Fig. 5 | Anti-MPXV antibodies trigger C1q and C3 deposition to block viral infection. a** Neutralization kinetics of VACV IHDJ by anti-MPXV mAbs at $IC_{50}$ concentrations with 1.5% NHS (3% during pre-incubation). NHS and mAbs were added at 10-min intervals from 2 h before to 2 h after infection. **b** Neutralization in the presence of 1.5% NHS (3% during pre-incubation) and 6 consecutive 4-fold dilutions of C5-IN-1 starting from 10 μM. **c** Same as (**b**) but with 8 consecutive 5-fold dilutions of a C3 inhibitor AMY-101 starting from 10 μM. **d** Neutralization in the presence of 1.5% NHS (3% during pre-incubation) or C1q- or C3-depleted human serum. **e** C1q deposition on VACV IHDJ virions (~5 × 10⁶ PFU) incubated with 1.5% NHS and 10 μg/mL mAbs. Left panel: flow cytometry plots, pre-gated for virions using GFP followed by staining for C1q-AF647. Right panel: median fluorescence intensity. **f-h** TEM visualization of mAb and complement factors C1q, and C3 deposition on virions. Purified MV particles were incubated with 10 μg/mL of either mAb 7M1162 anti-H3 (**f**), 8M2110 anti-A28 (**g**) or MGO53 isotype control (**h**) and 1.5% NHS. IgG, C1q, and

C3 deposition were visualized using anti-human IgG gold nanoparticles (10 nm) or secondary mouse anti-C1q and mouse anti-C3, followed by labeling with anti-mouse gold nanoparticles (20 nm). Anti-H3 and anti-A28 mAbs are shown in magenta and green, respectively; anti-M1 (7D11) in black; MGO53 in gray. $IC_{50}$ concentrations for panels (**a–d**) were: 15 μg/mL (7M1162), 0.125 μg/mL (8M2110), 1.25 μg/mL (10M2146), and 62.5 μg/mL (10M2154). Spearman correlation was used in (**a**) to determine the relationship between pre-incubation times and neutralization. Standard curves were determined by fitting values using simple linear regression for (**a**) and Inhibitor vs. normalized response (Variable slopes) nonlinear regression for (**b-c**). Standard deviation of the mean is shown for all values in (**a–d**). The percentages of complement sources indicated represent the final concentrations in the infected wells. Data are representative of two independent experiments with similar results using technical replicates: n = 3 for (**a**), n = 5 for (**b**), n = 6 for (**c**), and n = 3 for (**d**). Source data are provided as a Source Data file.

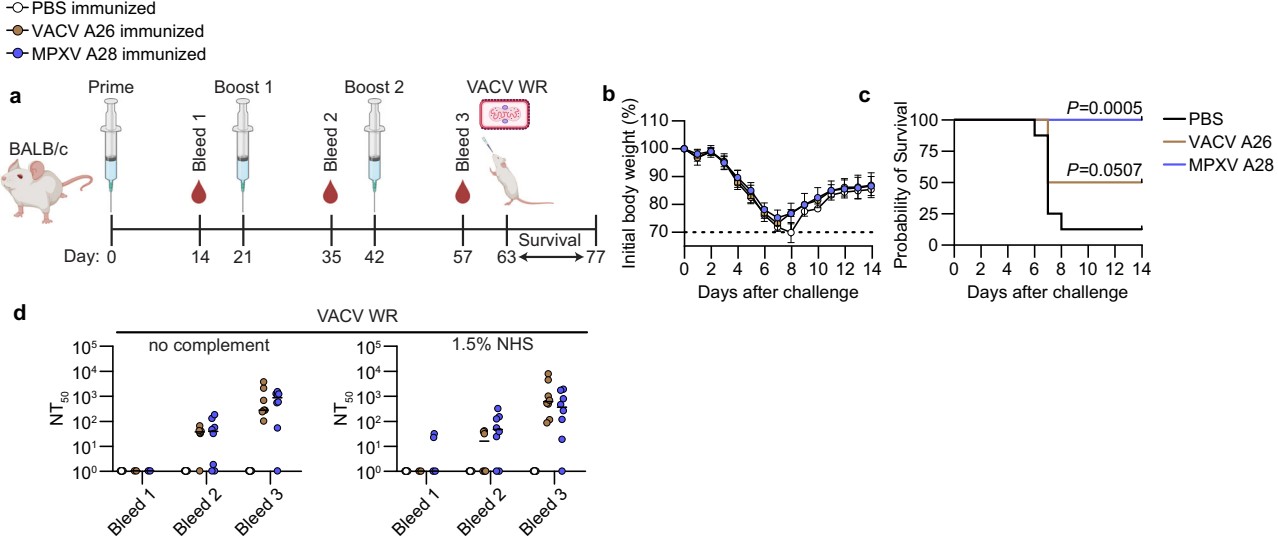

**Fig. 6 | Antibodies against A28 confer protection from a lethal VACV challenge.** **a** Schematic representation of mice A28 immunization and challenge. Female BALB/c mice ($n = 8$ mice per group) were immunized I.P. with either PBS, MPXV A28, or VACV A26 antigens (10 µg per dose), formulated with QS-21 adjuvant. Immunizations were administered three times (prime + two boosts) at 3-week intervals. Two weeks after each injection, mice were bled and serum was collected and heat inactivated. Three weeks after the final boost, mice were challenged I.N. with a lethal dose of VACV WR ($2 \times 10^5$ PFU). Weight and survival were monitored daily for 14 days. Mice that lost more than 30% of their initial weight (or more than 25% and exhibited a core body temperature below 34 °C) were considered to have reached the no recovery threshold and were subsequently sacrificed. **b** Body weight changes relative to starting weight for A28-, A26-, or PBS-immunized mice. Values are shown as mean percent weight loss ± standard deviation for each group ($n = 8$ mice per group). **c** Kaplan-Meier survival curve of A28-, A26- or PBS- immunized mice. **d** Time course analysis of heat-inactivated sera ($n = 8$ mice per group) neutralization of VACV WR strain with or without 1.5% NHS (3% during pre-incubation). $NT_{50}$ values were calculated from neutralization curves obtained through six consecutive 4-fold dilutions, starting at 1:40. Left panel: neutralization in the absence of NHS. Right panel: neutralization in the presence of 1.5% NHS (3% during pre-incubation). PBS-immunized mice are in white, VACV A26-immunized mice are in brown, and MPXV A28-immunized mice are in blue. In panel (**c**), statistical analysis was conducted by comparing survival curves to the isotype control using the Log-rank (Mantel-Cox) test, with Holm-Sidak multiple comparison correction applied in (**c**). The percentages of complement sources indicated represent the final concentrations in the infected wells. Images were created using *BioRender. Freund, N.*(https://BioRender.com/y6l0xzo). Data are representative of two independent experiments with similar results (**d**). Source data are provided as a Source Data file.

controlled through immunization, culminating in the successful eradication of Smallpox in 1980[64]. While the non-replicating MVA vaccine strain mitigated viral spread[8], its antigenic makeup differs from MPXV[49,65], and the immune response rapidly declines over time[9–11]. The low immunogenicity of MVA, along with recent advances in vaccine technology, has facilitated the development of new antigen-based vaccines[66–69]. Recent studies have demonstrated that poxvirus mRNA- and subunit-based vaccines elicit robust neutralizing and protective antibody responses in both mouse and non-human primate (NHP) models[33,66–71]. These vaccine candidates primarily include antigens previously described to be targets of neutralizing antibodies and therefore exclude A28 as an immunogen. Including A28 in such formulations could enhance the resilience of the immune response, particularly given the extensive APOBEC3-driven mutational changes observed in MPXV, which could lead to antigenic escape in other vaccine components[72,73]. Moreover, our findings indicate that antibodies against A28 are largely germline-encoded, requiring minimal affinity maturation, suggesting they can be readily induced by vaccination. We show that immunization of mice with recombinant A28 elicits strong antigen-specific B cell responses and neutralizing antibodies, some of which functioned independently of complement. The superior protection observed with vaccination compared to mAb treatment may stem from a broader polyclonal response that targets additional epitopes beyond those recognized by the mAbs. It may also reflect properties of the immunogen or enhanced Fc-mediated functions acting in synergy with the immune system. These mechanisms require further investigation but support the relevance of A28 as a promising subunit for next-generation Poxvirus vaccines.

## Limitations of the study

This study has several limitations. Due to the complexity of the MPXV and its abundance of surface proteins, we focused on a limited number of antigens (A35, H3, and A28) to assess the immune response, potentially overlooking other relevant targets that contribute to protection. A broader analysis including additional viral antigens could provide a more comprehensive understanding of the antibody response. Second, while we employed antigen-specific B cell sorting, some of the antigens used in our study, particularly H3 and A28, also serve as viral attachment proteins. This likely led to non-specific binding to B cells during sorting and may explain the small proportion of antigen-binding mAbs. As a result, we were unable to fully characterize the B cell repertoire responding to these antigens, potentially leading to an incomplete representation of the antibody diversity generated post-infection. Moreover, most of the neutralization assays performed in this study were MV-focused, which may underestimate the contribution of anti-A35 mAbs targeting EV particles to overall neutralization. These anti-A35 mAbs will be investigated in follow up studies. Additionally, due to biosafety level restrictions, we primarily used VACV as a surrogate to study the mechanism of mAb neutralization rather than MPXV. While VACV shares many structural and functional similarities with MPXV, differences between the two viruses may limit the direct applicability of our findings to MPXV-specific immunity.

## Methods
### Cell lines and viruses
Expi293F cells (Thermo Fisher Scientific, CAT#A14527) were grown in Expi293 Expression Medium (Thermo Fisher Scientific) at a constant shaking speed of 110 RPM and maintained at 37 °C with 8% $CO_2$. HeLa

(human female cervical cancer, ATCC CCL-2), Vero (African green monkey kidney epithelial cells, ATCC CCL-81), and U2OS (human female osteosarcoma cells, ATCC HTB-96) cells were cultured in DMEM (Sigma-Aldrich) supplemented with 10% heat-inactivated (HI) FBS (Hyclone), 100 U/mL penicillin, 100 U/mL streptomycin, and 2 mM L-glutamine (Thermo Fisher Scientific). Cells were maintained at 37 °C with 5% $CO_2$.

The VACV IHDJ strain used for ELISA was kindly provided by Prof. Ehud Katz and was inactivated using β-propiolactone (βpL) by the Israel Institute of Biological Research (IIBR) prior to use under BSL-1 conditions. VACV-vFIRE-WR, VACV-vFIRE-Copenhagen, and VACV-vFIRE-IHDJ were kindly provided by Prof. Bernard Moss. The MPXV sub-clade IIb (MPXV/2022/FR/CMIP) and MPXV sub-clade Ib (MpxV/PHAS-506/Passage- 03/SWE/2024_09_11) were isolated and propagated as previously described[46,74,75].

VACV-vFIRE-WR, VACV-vFIRE-Copenhagen, and VACV-vFIRE-IHDJ were propagated on a confluent monolayer of HeLa cells in MEM (Gibco) supplemented with 2% HI FBS, 100 U/mL penicillin, 100 U/mL streptomycin, 2 mM L-glutamine, and 1% non-essential amino acids (NEAAs) (Sartorius). After 2 h, the cells were overlaid with MEM supplemented with 8% HI FBS, 100 U/mL penicillin, 100 U/mL streptomycin, 1% NEAAs, and 2% sodium bicarbonate solution (Merck). After 72 h of virus propagation, infected cells were scraped from the tissue culture dish and pelleted by centrifugation at $1800 \times g$ for 5 min. Viruses were harvested by three freeze-thaw cycles, and cell debris was removed by centrifugation at $1800 \times g$ for 5 min. Viruses were titrated on a confluent monolayer of Vero cells. Briefly, Vero cells were infected with serially diluted virus. After 1 h of infection, the cells were overlaid with DMEM containing methylcellulose (Sigma-Aldrich) and incubated for 72 h, followed by plaque visualization using Coomassie staining (Bio-Rad). For the immunogold labeling and transmission electron microscopy experiments, purified MV particles, infected cell lysates were centrifuged and overlaid onto a 36% sucrose cushion, followed by ultracentrifugation at $33,000 \times g$ for 80 min. The resulting pellet was then resuspended in 10 mM Tris-HCl (pH 9).

All experiments with infectious VACV strains were performed under BSL-2 or ABSL-2 and conditions. All experiments with infectious MPXV were performed under strict BSL-3 conditions. In all assays using infectious MPXV or VACV, the viral particles used were MV-enriched preparations obtained by three freeze–thaw cycles, without further purification, unless otherwise specified.

## Mice

Female C57BL/6JOlaHsd and BALB/cOlaHsd mice were purchased at 7 weeks of age from Harlan Laboratories. Animals were housed in ventilated microisolator cages under standard conditions, including controlled temperature ($22 \pm 2$ °C), relative humidity ($50 \pm 10$%), and a 12-h light/dark cycle, with 4–5 animals per cage. Bedding was provided and regularly changed, and animals had ad libitum access to food and water. Environmental enrichment, such as nesting material and shelters, was provided. All animal experiments were conducted under strict ABSL-2 conditions, and only vaccinated, authorized researchers handled the animals, following institutional guidelines for biosafety and animal welfare. Handling and procedures were performed to minimize stress and discomfort, in accordance with approved animal protocols. Experimentation complied with the Tel Aviv University Institutional Animal Care and Use Committee (permit numbers TAU-MD-IL-2407-149-4, TAU-MD-IL-2411-181-5, and TAU-MD-IL-2502-104-5).

## Sample collection and processing

Peripheral whole blood samples were collected at 1–2 months (time point 1, V1) and 9–10 months (time point 2, V2) following PCR-confirmed Mpox infection. Serum was isolated from 5 mL of each sample, while the remaining whole blood was used for peripheral blood mononuclear cell (PBMC) isolation using a Ficoll gradient

(Cytiva), as previously described[37]. For serum isolation, whole blood was allowed to clot and then centrifuged at $2000 \times g$ for 15 min, followed by heat inactivation at 56 °C for 20 min.

## Expression of MPXV antigens and VACV homologs

Expression of recombinant MPXV antigens A35 and H3 was previously described[18]. MPXV (GenBank: MN648051)[76] and VACV IHDW strain (GenBank: KJ125439)[77] sequences were used as templates for the production of MPXV antigen A28 (OPG 153) and the VACV homologs A33 (OPG 161), H3 (OPG 108), and A26 (OPG 153). The amino acid regions expressed for each antigen were as follows: MPXV A28 (2–361), VACV A33 (58–185), VACV H3 (2–278), and VACV A26 (2–361). Construct design was similar to that of MPXV antigens A35 and H3[18]. Briefly, antigen amino acid sequences were codon-optimized for mammalian expression and modified to include a growth factor receptor signal peptide at the N-terminus. An 8×Histidine (His-tag) sequence followed by a biotinylation encoding sequence (Avi-tag) was added at the C-terminus. Antigen-encoding DNA sequences were synthesized by an external vendor, cloned into the pcDNA3.1 (-) vector, and transiently transfected into Expi293F cells following the manufacturer's protocol. Seven days post-transfection, cell supernatants were incubated with nickel beads (Cytiva) overnight at 4 °C. The beads were then loaded onto chromatography columns (Bio-Rad), washed, and the antigens were eluted using increasing concentrations of imidazole (Thermo Scientific), followed by dialysis against PBS overnight. For experiments requiring biotinylated antigens, buffer exchange was performed with 10 mM Tris-HCl, and biotinylation was carried out using the BirA kit (Avidity), following the manufacturer's protocol. Biotinylated antigens were then buffer-exchanged back to PBS.

MPXV H3 point mutations were generated by PCR amplification of the MPXV H3 plasmid with corresponding nucleotide substitutions and cloned as previously described[18]. H3 mutations were verified by Sanger sequencing, and the mutant H3 proteins were produced in Expi293F cells as described above.

## Western blot analysis of purified antigens

Validation of proper expression of MPXV and VACV antigens was performed using SDS-PAGE and Western blotting. Briefly, 0.5 μg of each antigen, containing 25 mM DTT (Sigma-Aldrich), along with an unstained protein standard (Bio-Rad), were loaded onto an SDS-PAGE and transferred to a nitrocellulose membrane (Bio-Rad). The membrane was blocked for 2 h at room temperature (RT) in blocking buffer containing PBS, 3% BSA (ENCO), 0.05% Tween-20 (Sigma-Aldrich), and 20 mM EDTA (Bio-Labs). Anti-Avi-Tag mouse antibody (Avidity), diluted 1:5000 in blocking buffer, was incubated with the membrane overnight at 4 °C. After 5 washes with washing buffer containing PBS and 0.05% Tween-20, the membrane was incubated for 45 min at RT with a secondary anti-mouse HRP-conjugated antibody (Jackson Laboratory) and StrepTactin-HRP (Bio-Rad), both diluted 1:5000 in blocking buffer. Following 5 additional washes, ECL substrate was added, and Western blot images were acquired in Chemidoc imaging system (Bio-Rad).

## Antigen-binding single B cell sorting and sequencing

PBMCs from Mpox convalescent donors were thawed in RPMI 1640 (Sigma-Aldrich) medium warmed to 37 °C, followed by centrifugation and resuspension in MACS buffer containing PBS, 0.5% BSA, and 2 mM EDTA. B cells were then enriched using anti-CD19 (Miltenyi Biotec) magnetic beads according to the manufacturer's protocol, followed by centrifugation and resuspension in FACS buffer containing PBS, 1% FBS, and 2 mM EDTA for subsequent staining. The B cell-enriched samples were stained with Labeling Check Reagent-VioBlue (1:100; Miltenyi Biotec), IgG-FITC (1:100; BioLegend), and either biotinylated MPXV antigens A35, H3, or A28, labeled with both Streptavidin-PE (1 μl per 1 μg of protein; BioLegend) and Streptavidin-Alexa Fluor 647 (1 μl

per 1 μg of protein; BioLegend), for 30 minutes at 4 °C. After washing and resuspending the cells in FACS buffer, CD19⁺IgG⁺antigen⁺ cells were single-cell sorted using the FACSAriaIII sorter into 96-well PCR plates containing 4 μL of lysis buffer (PBS×0.5, 12 units of RNasin Ribonuclease inhibitor (IMBH), and 10 mM DTT (Invitrogen), and immediately frozen on dry ice. Plates were thawed on ice for subsequent cDNA synthesis and PCR amplification as previously described[37,78]. Briefly, lysed cells were prepared for reverse transcription by adding random hexamer primers (Invitrogen), IGEPAL (Sigma-Aldrich), and RNasin Ribonuclease Inhibitor to the sorted wells and incubating at 68 °C for 1 min. cDNA was then synthesized using a reverse transcription PCR reaction containing SuperScript III reverse transcriptase (Invitrogen), dNTPs (Thermo Scientific), DTT, and RNasin Ribonuclease Inhibitor. The reaction was carried out under the following conditions: 42 °C for 10 min, 25 °C for 10 min, 50 °C for 60 min, and 94 °C for 5 min. Assembled cDNA was used as a template for subsequent $Ig_H/Ig_L$ amplification by two-step nested PCR. The first round of PCR was performed using Phusion High-Fidelity DNA Polymerase (NEB), dNTPs, and the corresponding immunoglobulin Gamma, Kappa, or Lambda primer mix[37,78]. The reaction conditions were as follows: initial denaturation at 98 °C for 30 seconds, followed by 50 cycles of denaturation at 98 °C for 10 seconds, annealing at 52 °C (Gamma), 50 °C (Kappa), or 58 °C (Lambda) for 15 seconds, extension at 72 °C for 15 seconds, and final extension at 72 °C for 5 min. The second PCR was performed under similar conditions, but the number of cycles was reduced to 40, and the annealing temperatures were adjusted to 56 °C (Gamma), 50 °C (Kappa), and 60 °C (Lambda). To verify DNA amplification, the second-round PCR products were visualized on a 2% agarose gel and sent for Sanger sequencing.

The retrieved $Ig_H/Ig_L$ sequences were analyzed and annotated using IgBLAST and the IMGT database for sequence quality, viability, Gamma chain subtype ($Ig_H$ only), $V_H D_H J_H$ or $V_L J_L$ gene usage, mutational load, and CDR3 length. Cells displaying identical V and J gene usage and ≥70% homology in both $Ig_H/Ig_L$ sequences were classified as clonally expanded B cells. Germline sequences for each mAb were determined using the IMGT database. For $Ig_H$ CDR3, the germline sequence was set as the retrieved $Ig_H$ CDR3 sequence itself. In cases where clonally expanded B cells were identified, both CDR3 $Ig_H/Ig_L$ sequences were determined by assessing the most likely common features based on their $D_H$ IMGT sequence.

## Anti-MPXV antibody cloning and production

Viable $Ig_H/Ig_L$ sequences recovered from previous steps were selected for production as IgG1 mAbs. Briefly, first-round PCR products were used as templates for an additional round of PCR amplification with specific 5' and 3' primers encoding restriction sites for subsequent cloning. This additional PCR was performed under conditions similar to the previously described second-round PCR, with an annealing temperature of 56 °C for all reactions. PCR products were visualized on a 2% agarose gel, purified, digested with the appropriate restriction enzymes (NEB), and ligated into the relevant immunoglobulin expression vectors: IgG1 (Gamma), IgK (Kappa), and IgL (Lambda). The corresponding heavy (Gamma) and light (Kappa or Lambda) chain plasmids were transiently co-transfected at a 1:3 ratio into Expi293F cells following the manufacturer's protocol. Seven days post-transfection, the supernatant was incubated with protein A beads (Cytiva) for 2 h at room temperature (RT). Beads were then loaded onto chromatography columns, washed, and mAbs were eluted using a low-pH solution before being dialyzed against PBS overnight. For experiments requiring biotinylated mAbs, biotinylation was performed using EZ-Link Sulfo-NHS-LC-Biotin (Thermo-Scientific) according to the manufacturer's protocol, followed by an additional overnight dialysis against PBS. To produce germline-reverted mAbs, codon-optimized sequences of the germline-reverted mAbs (as described above) were synthesized by an external vendor and cloned into the relevant immunoglobulin expression vector. Plasmids were transiently co-transfected into Expi293F cells, and germline-reverted mAbs were isolated as previously described.

Anti-VACV-L1 mouse recombinant mAb (7D11)[79] was used as a positive control for subsequent neutralization assays, as previously described[26,68]. Briefly, the $Ig_H/Ig_L$ sequences of 7D11 were obtained from the Protein Data Bank (PDB: 2I9L)[80]. The corresponding amino acid sequences were codon-optimized, synthesized by an external vendor, and cloned into a human IgG1 and IgK immunoglobulin expression vectors. Cloned plasmids were transiently co-transfected into Expi293F cells, and the humanized 7D11 mAb was purified as previously described.

## Enzyme-linked immunosorbent assay

**VACV IHDJ ELISA.** βpL-inactivated VACV IHDJ strain was diluted in carbonate-bicarbonate solution to a concentration of $10^7$ PFU/mL and used to coat high-binding ELISA plates (Corning) overnight at 4 °C. Plates were then blocked for 2 h at RT with blocking buffer comprising PBS, 3% BSA, 0.05% Tween-20, and 20 mM EDTA, followed by a single wash with washing buffer comprising PBS and 0.05% Tween-20. Anti-MPXV mAbs were serially diluted 4-fold in blocking buffer, beginning at 100 μg/mL, for a total of four dilutions before being added to the plates for 1 h at RT. After three washes with washing buffer, plates were incubated for 45 min at RT with an anti-human IgG HRP-conjugated antibody (Jackson Laboratory) diluted 1:5000 in blocking buffer. Following five additional washes, TMB/E (Abcam) was added, and optical density (O.D.) was measured at 650 nm after 10 min.

**MPXV and VACV antigen ELISA.** 2.5 μg/mL of antigen was used to coat high-binding ELISA plates overnight at 4 °C. Plates were then blocked for 2 h at RT with a blocking buffer, followed by a single wash with washing buffer. For the Mpox convalescent and vaccinated donor ELISA, HI serum samples were serially diluted 4-fold in blocking buffer, starting at 1:100, for a total of four dilutions. Alternatively, for anti-MPXV mAb ELISA, mAbs were diluted in blocking buffer across 12 consecutive 4-fold dilutions, starting at 10 μg/mL. Serum or mAb samples were added to the ELISA plates and incubated for 1 h at RT. After three washing cycles with washing buffer, plates were incubated with an anti-human IgG HRP-conjugated antibody (diluted 1:5000 in blocking buffer) for 45 min at RT. After five additional washes, TMB/E was added, and the O.D. was measured at 650 nm after 10 min.

**Immunized mice ELISA.** 2.5 μg/mL of antigen was used to coat high-binding ELISA plates overnight at 4 °C. Plates were then blocked for 2 h at RT with a blocking buffer, followed by a single wash with washing buffer. HI vaccinated mouse serum was serially diluted 4-fold in blocking buffer, starting at 1:100 and added to the ELISA plates and incubated for 1 h at RT. After three washing cycles with washing buffer, plates were incubated with an anti-mouse IgG HRP-conjugated antibody (Jackson Laboratory) (diluted 1:5000 in blocking buffer) for 45 min at RT. After five additional washes, TMB/E was added, and the O.D. was measured at 650 nm after 10 min.

**Anti-MPXV mAbs competition ELISA.** 1 μg/mL of MPXV A35 and H3, and 2.5 μg/mL of MPXV A28, were used to coat high-binding ELISA plates overnight at 4 °C. The plates were then blocked for 2 h at RT in blocking buffer, followed by a single wash with washing buffer. The anti-MPXV mAbs were diluted in blocking buffer across 8 consecutive 4-fold dilutions, starting at 64 μg/mL, and added to the ELISA plates for 30 min at RT. Next, 0.1 μg/mL of biotinylated mAbs (4M1130, 4M1133, or 4M1154 for A35-coated plates, 5M1111 or 7M1162 for H3-coated plates, and 8M2110 or 10M2146 for A28-coated plates) were added to each well and incubated for an additional 15 min. The plates were then washed three times with washing buffer and incubated with Streptavidin-HRP (Jackson Laboratory), diluted 1:5000 in blocking

buffer, for 45 min at RT. After 5 additional washes, TMB/E was added, and the O.D. was measured at 650 nm after 10 min.

**Affinity-selected phage ELISA.** 2 µg/mL of mAb 7M1162 was used to coat high-binding ELISA plates overnight at 4 °C. The plates were then blocked for 2 h at RT in a blocking solution comprising TBS, 5% skim milk (Difco), and 0.05% Tween20, followed by a single wash with washing solution comprising TBS and 0.05% Tween20. Corresponding affinity-selected phages were diluted across 8 consecutive 4-fold dilutions in blocking solution, starting from $4 \times 10^8$ phages per well, and added to the ELISA plates for 1 h at RT. The plates were then washed three times with washing solution and incubated with rabbit anti-M13 antibody diluted 1:5000 in blocking solution for 1 h at RT. After 3 additional washes, the plates were incubated with anti-rabbit IgG HRP-conjugated antibody (Merck) diluted 1:5000 in blocking solution for 45 min at RT. Following 5 more washes, TMB/E was added, and the O.D. was measured at 650 nm after 10 min.

## Virus neutralization assays

**VACV strains.** WR (VACV-vFIRE-WR), Copenhagen (VACV-vFIRE-Copenhagen), and IHDJ (VACV-vFIRE-IHDJ), as well as MPXV clade IIb (MPXV/2022/FR/CMIP) and MPXV clade Ib (MpxV/PHAS-506/Passage-03/SWE/2024_09_11), were used for antibody neutralization assays. All assays were performed on U2OS cells, seeded at $2 \times 10^4$ cells per well in 96-well plates 24 h prior to infection, using MV-enriched viral preparations as described above.

**VACV WR, Copenhagen, and IHDJ strains neutralization assay.** Anti-MPXV mAbs or HI mouse sera were diluted in 6 consecutive 4-fold dilutions in PBS, starting from 250 µg/mL and 1:10, respectively. For experiments requiring $IC_{50}$ determination, mAbs were diluted in 12 consecutive 4-fold dilutions. Antibodies were mixed at a 1:1 ratio with cell medium containing $4 \times 10^3$ PFU/mL of either VACV WR, Copenhagen, or IHDJ in the presence or absence of 6% Normal Human Serum (NHS- Sigma-Aldrich) or HI NHS. After a 2-h incubation at 37 °C, the virus inoculum was added to the U2OS cell monolayer at a 1:1 ratio. The virus-infected wells had a final virus concentration of ~100 PFU/well (MOI of 0.005), 1.5% NHS, 4-fold dilutions of mAbs starting from 62.5 µg/mL, or 4-fold dilutions of mouse sera starting from 1:40. For experiments in which anti-MPXV mAbs were tested in combinations, mAbs were mixed at a 1:1 ratio between themselves, reaching a final IgG concentration of 62.5 µg/mL (31.25 µg/mL of each mAb- final concentration) and subsequently serially diluted in PBS if needed. Following incubation for 48 h at 37 °C and 5% CO2, plates were sealed with breathable sealing tape (Thermo Scientific), and images were acquired using the IncuCyte live-cell analysis system. Virus infection levels were quantified by analyzing the GFP area in each well.

**MPXV neutralization assay.** The neutralization assays using MPXV sub-clades Ib and IIb were performed as previously described[46,75]. Anti-MPXV mAbs were diluted 6 consecutive 4-fold dilutions in PBS, starting from 250 µg/mL. For experiments requiring $IC_{50}$ determination, mAbs were diluted 8 consecutive 4-fold dilutions starting from 100 µg/mL. MAbs were mixed at a 1:1 ratio with cell medium containing MPXV in the presence or absence of 40% Guinea pig complement (GPC-Rockland) or 4% NHS. After 2 h of incubation at 37 °C, virus inoculum was added onto the U2OS cells monolayer at a 1:1 ratio. The virus-infected wells had a final concentration of 10% GPC or 1% NHS, and 6 or 8 consecutive 4-fold dilutions of mAbs, starting from 62.5 µg/mL or 25 µg/mL, respectively. For experiments in which anti-MPXV mAbs were tested in combinations, mAbs were mixed at a 1:1 ratio between themselves, reaching a final IgG concentration of 62.5 µg/mL (31.25 µg/mL of each mAb- final concentration). Following incubation for 48 h at 37 °C and 5% CO2, cells were fixed for 30 min with 4% PFA (Invitrogen), washed, and stained with rabbit polyclonal anti-VACV antibodies

(1:6000; Invitrogen). Following this, plates were stained with an anti-rabbit antibody coupled to Alexa Fluor 488 (1:400; Invitrogen) and washed with Hoechst (1:10000; Invitrogen) diluted in PBS. Images were acquired with an Opera Phenix high-content confocal microscope, and virus infection levels were quantified by analyzing the fluorescence area in each well, as previously described[46].

**Neutralization kinetics assay.** VACV IHDJ neutralizing mAbs were used in their $IC_{50}$ values: 15 µg/mL for 7M1162, 0.125 µg/mL for 8M2110, 1.25 µg/mL for 10M2146, and 62.5 µg/mL for 10M2154 in the presence of 1.5% NHS (final concentrations in the virus-infected wells). All wells were infected with 100 PFU/mL of VACV IHDJ simultaneously. The addition of diluted mAb and NHS were performed in 10 min intervals, starting 2 h pre-infection and lasting until 2 h post-infection. For pre-infection wells, mAbs and NHS were mixed in wells containing cell medium and ~100 PFU/well of VACV IHDJ prior to its infection of the U2OS cells. For post-infection wells, mAbs and NHS were added directly to the previously infected cells. Analysis of neutralization was performed similarly to VACV neutralization assays described above.

**Complement inhibitors and depleted sera neutralization assays.** VACV IHDJ neutralizing mAbs were used at their $IC_{50}$ values: 15 µg/mL for 7M1162, 0.125 µg/mL for 8M2110, 1.25 µg/mL for 10M2146, and 62.5 µg/mL for 10M2154. For experiments requiring complement depleted sera, mAbs were incubated in the presence of 1.5% NHS, C1q-, C3-depleted sera (Complement Technology, Inc., final concentrations in the virus-infected wells). For experiments involving selective complement inhibitors, mAbs and NHS mixtures were incubated with either 6 consecutive 4-fold dilutions, starting from 10 µM of a selective complement C5 inhibitor, C5-IN-1 (Compound 7- MedChemExpress), or 8 consecutive 5-fold dilutions, starting from 10 µM of a selective complement C3 inhibitor, CP40 (AMY-101- MedChemExpress). Analysis of neutralization was performed similarly to VACV neutralization assays described above.

**Exogenous C1q neutralization assays.** VACV IHDJ neutralizing mAbs were used at their $IC_{50}$ values: 15 µg/mL for 7M1162, 0.125 µg/mL for 8M2110, 1.25 µg/mL for 10M2146, and 62.5 µg/mL for 10M2154 in the presence or absence of 1.2 µg/mL of human purified C1q, representing the amount found in 1.5% NHS (final concentrations in the virus-infected wells). Analysis of neutralization was performed similarly to VACV neutralization assays described above. The level of infection in samples containing C1q was normalized to those without.

In all virus neutralization assays, except where noted otherwise, the percentage of neutralization was calculated based on the reduction of fluorescence area (GFP for VACV or Alexa Fluor 488 for MPXV). Due to infection variability between wells, to ensure antibody-treated wells show genuine fluorescence area reduction, all non-antibody-treated wells were analyzed for their fluorescence area values. Baseline infection for each experiment was set as the non-antibody-treated well displaying the least fluorescence area (baseline infection). The percentage of neutralization for each well was calculated using the following formula: 100 - (fluorescence area value / baseline infection fluorescence area) × 100. Neutralization rates calculated as negative values were set to zero. All viral preparations used in these assays were enriched for MV particles, as described above.

## VACV infected cells binding assay

Confluent monolayers of Vero cells were infected with VACV IHDJ (MOI of 1) for 14 h until GFP expression became evident. Infected cells were then detached from the tissue culture dish using Trypsin/EDTA (Thermo Scientific), centrifuged at $400 \times g$ for 5 min, and resuspended in FACS buffer containing 10 µg/mL of the anti-MPXV mAbs. The suspension was incubated for 30 min at 4 °C. Following incubation, cells were washed, centrifuged again at $400 \times g$ for 5 min, and resuspended

in FACS buffer containing anti-IgG-Alexa Fluor 647 (Alexa Fluor 488 (1:100; BioLegend) for an additional 30 min at 4 °C. Labeled cells were washed one more time and fixed with 4% PFA for 20 min at RT. After a final wash, the labeled cells were analyzed using a Flow Cytometer.

## VACV infected cells CDC assay

U2OS cells were seeded at $2 \times 10^5$ cells per well in a 6-well plate and incubated at 37 °C and 5% CO2 for 24 h until reaching ~50% confluency. The cells were then infected with VACV IHDJ (MOI of 0.1) for an additional 24 h, during which GFP expression became evident. To remove free viral particles, infected cells were washed twice with fresh media containing 1:1000 Propidium iodide (PI- BioLegend). Plates were sealed with breathable sealing tape, and the number of dead/total cells was analyzed using the IncuCyte live-cell analysis system at time point 0. After image acquisition, 10 µg/mL of anti-MPXV mAbs in the presence of 1.5% NHS were added to the wells containing infected cells. Plates were re-sealed, and further analysis of dead/total cells was conducted using the IncuCyte system. Images were acquired over the next 24 h at 1-h intervals.

## VACV C1q deposition

MV-enriched purified VACV IHDJ virions ($\sim 5 \times 10^6$ PFU) were incubated with 62.5 µg/mL of anti-MPXV mAbs and 1.5% NHS for 2 h at 37 °C. Following incubation, the virions were stained with anti-C1q-Alexa Fluor 647 (1:100; Santa Cruz) for 1 h. PFA was then added to the virions to reach a final concentration of 4%, followed by analysis of the GFP-expressing virions in the Flow Cytometer.

## Immunogold labeling and transmission electron microscopy

Carbon/Formvar-coated copper grids (Bar Naor) were placed on a 40 µL drop of purified VACV IHDJ MV ($5 \times 10^8$ PFU/mL) and incubated at room temperature for 15 min to facilitate virus adsorption. The grids were washed once with PBS and subsequently incubated in blocking solution (3% BSA in PBS) for 30 min. Following blocking, grids were incubated overnight at 4 °C with 10 µg/mL of anti-MPXV mAbs in the presence of 1.5% NHS in blocking solution. The next day, grids were transferred to 37 °C to allow complement activation, followed by two PBS washes and a 10-min incubation in blocking solution. For immunogold labeling, grids were incubated for 3 h at room temperature with either anti-C1q (1:200; Santa Cruz) or anti-C3 (1:200; BioLegend), followed by additional washing and blocking steps. Secondary labeling was performed using anti-human IgG gold nanoparticles (10 nm-Abcam; diluted 1:50 in blocking solution) to visualize anti-MPXV mAbs and anti-mouse gold nanoparticles (20 nm- Abcam; diluted 1:50 in blocking solution) for C1q or C3 detection. Unbound antibodies were removed by two consecutive PBS washes (10 min each). Samples were fixed with 1% glutaraldehyde (Sigma-Aldrich) in PBS, rinsed with ultrapure water, and negatively stained with Uranyless (Electron Microscopy Sciences) according to the manufacturer's instructions. The grids were air-dried and imaged using a JEM-1400 transmission electron microscope (JEOL Ltd, Tokyo, Japan) operating at 80 kV.

## Mice immunization

**Passive immunization.** Female BALB/cOlaHsd (BALB/c) mice (8 weeks old) received 200 µg of either monoclonal antibody 8M2110 (anti-A28) or the isotype control MGO53, administered intraperitoneally in PBS. The following day mice were challenged.

**Active immunization.** For VACV challenge experiments, female BALB/cOlaHsd (BALB/c) mice (8 weeks old) were immunized intraperitoneally with 10 µg of either MPXV A28, VACV A26, or PBS, formulated with 1.5 µg of QS-21 adjuvant (InvivoGen) per mouse. Immunizations were administered three times (prime plus two boosts) at 3-week intervals. Two weeks after each injection, blood was collected and sera was heat inactivated. Three weeks after the final boost, mice were

challenged. For immune cell profiling, female C57BL/6 J mice (8 weeks old) were immunized intraperitoneally with the same antigen formulations (10 µg per dose), adjuvanted with Alum (Thermo Scientific) in a 1:1 volume ratio (final volume 200 µL). Immunizations were administered on days 1, 26, and 56. Mice were bled one day before each dose, and sera were collected and heat inactivated. On day 76, mice were sacrificed, and blood and organs were harvested for downstream analyses.

## VACV challenge experiments

BALB/cOlaHsd (BALB/c) mice, either from the passive or active immunization experiments were housed under ABSL-2 conditions. Mice were anesthetized using Ketamine/Xylazine and inoculated I.N. with a lethal dose of VACV WR ($2 \times 10^5$ PFU). Weight and survival were monitored daily for 14 days. Mice that lost more than 30% of their initial weight (or more than 25% and exhibited a core body temperature below 34 °C) were considered to have reached the no recovery threshold and were subsequently sacrificed. For viral load analysis, mice were inoculated with VACV WR as described above and sacrificed on day 5 post-infection.

## Splenocytes immune cell analysis

Flow cytometry was used to analyze immune cell populations in spleens collected from vaccinated (C57BL/6 JOlaHsd) C57BL/6J mice. On day 76, three weeks after the second boost, mice were euthanized and spleens were processed into single-cell suspensions, aliquoted, and stored frozen until further analysis, as previously described[40].

Frozen aliquots were thawed in RPMI 1640 medium pre-warmed to 37 °C, centrifuged, and resuspended in FACS buffer. Cells were stained for 30 min at 4 °C with either B220-PerCP-Cy5.5 (1:100; Invitrogen), GL7-Alexa Fluor 647 (1:100; BioLegend), FAS(CD95)-PE-Cy7 (1:100; Miltenyi Biotec), CD3-APC-Cy7 (1:100; BioLegend), CD4-BV421 (1:100; BioLegend), and CD8a-FITC (1:100; BioLegend), or with B220-PerCP-Cy5.5, IgG-BV605 (1:100; BioLegend), IgM-FITC (1:100; BioLegend), and MPXV A28 conjugated to both Streptavidin-PE and Streptavidin-Alexa Fluor 647 (1 µl per 1 µg of protein). Following a final wash, stained cells were analyzed by flow cytometry.

## Viral load

Lungs from infected mice were stored in PBS containing 0.1% BSA until homogenization using gentleMACS™ M Tubes and a gentleMACS™ Dissociator (Miltenyi Biotec). Viral load was quantified by titration on a confluent monolayer of Vero cells, as described above, and fluorescent foci were visualized and analyzed using the IncuCyte live-cell analysis system.

## X-ray crystallography

For structural studies, a synthetic gene encoding the head domain (amino acids 1–397) of A26 from the Vaccinia virus strain Western Reserve (Uniprot code: P24758), tagged at the C-terminus with a Strep-tag, was inserted into a pET-16b vector. The plasmid was transformed into *E. coli* BL21 (DE3) cells (New England Biolabs), and protein expression was induced overnight at 16 °C with 0.5 mM isopropyl β-D-1-thiogalactopyranoside (IPTG). Cells from 3 liters of culture were harvested, resuspended in 40 mL of cold lysis buffer supplemented with one tablet of complete protease inhibitor (Roche), and lysed by sonication. Insoluble material was removed by centrifugation at $20,000 \times g$ for 30 min at 4 °C. The recombinant protein was purified by affinity chromatography using a 5 mL StrepTrap™ HP column (Cytiva), followed by size-exclusion chromatography in TN8 buffer (10 mM Tris-HCl pH 8, 100 mM NaCl) on a Superdex 200 column (Cytiva) to remove aggregates. Purity was assessed by SDS-PAGE and the final protein yield was 2 mg/L, as determined by NanoDrop spectrophotometer. To produce full-length antibodies, Expi293 cells were seeded at $3 \times 10^6$ cells/mL and transfected in 200 mL cultures with 200 µg each of heavy

and light chain expression vectors using 200 μL of FectoPro transfection reagent. Cultures were incubated for 5 days at 37 °C in FreeStyle medium. Supernatants were clarified by centrifugation at 3100 × *g* for 30 min at 4 °C, followed by filtration through a 0.45 μm membrane. Antibodies were purified using a 5 mL Protein G column (Cytiva), eluted with 100 mM glycine pH 2.7, and further purified by size-exclusion chromatography in TN8 buffer on a Superdex 200 column. Purity was verified by SDS-PAGE, and concentrations were measured using a NanoDrop spectrophotometer. Final yields were approximately 30 mg/L for 10M2146 and 22 mg/L for 8M2110. Fab fragments of 8M2110 and 10M2146 were generated by papain digestion. For each antibody, 2.5 mg were incubated with 500 μL of 50% papain slurry (Thermo, cat. no. 20341). Fc fragments were removed using a Protein A column (Cytiva), and the monomeric Fab fraction was purified by size-exclusion chromatography.

For crystallization, purification tags were removed by overnight digestion with 1.5 units of thrombin (Cytiva) per 0.1 mg of protein at 4 °C. Complexes were assembled by incubating A26 with a molar excess of Fab (8M2110 or 10M2146) and an anti-Fab VHH that binds the constant region of the light chain and stabilizes the hinge region between the constant and the variable domains[81,82]. Complexes were purified by size-exclusion chromatography in TN8 buffer and concentrated to 3–5 mg/mL. Crystallization of the A26:8M2110:VHH complex was performed using 20% (w/v) PEG 3350 and 0.2 M ammonium citrate while the A26:10M2146:VHH complex was crystallized using 0.1 M MgCl₂, 0.1 M Tris-HCl pH 8.5, and 17% (w/v) PEG 20,000. In both cases, crystals were cryoprotected with ethylene glycol and diffraction data were collected at the SOLEIL synchrotron (St. Aubin, France) on beamlines PROXIMA-1 and PROXIMA-2. Diffraction data were processed using XDS (version January 10, 2022)[83], and initial phases were obtained using PHASER software[84] with the A26 model (PDB: 6A9S)[35] the VHH model (PDB: 7PIJ, chain E) and a model of the Fabs obtained using Alphafold3 as search templates. In both complexes, we observed clear electron density for all the components, and we built models comprising A26, the Fabs and the anti-Fab VHH (Supplemental Fig. 4). Structural models were built and refined iteratively using *phenix.refine* (Phenix version 1.19.2-4158)[85] and *Coot*[86], applying isotropic B-factor and one TLS group per chain. Model validation was performed with MolProbity[87]. Crystallographic statistics are provided (Supplemental Table 2). Coordinates and structure factors have been deposited in the Protein Data Bank under accession codes 9QT3 and 9SHD. Structural figures were generated with PyMOL v3.0.3 (Schrödinger, LLC).

## Biopanning and epitope prediction

Biopanning was performed as previously described[55]. Duplicate wells containing 2 μg/mL of anti-H3 mAb (7M1162) were incubated for 1 h with protein-G magnetic beads (Thermo Scientific) in a blocking buffer containing TBS×1 and 3% BSA. The antibody-bead mixture was placed on a magnetic stand, and the supernatant containing unbound mAbs was discarded. The beads were then washed three times with washing buffer (TBS×1 and 0.05% Tween-20) to remove non-specifically bound proteins. -10¹¹ phages from the random-peptide library were added to the antibody-bound magnetic beads in blocking buffer and incubated for 1 h. After incubation, the unbound phages were removed by discarding the supernatant, and the beads were washed three times with washing buffer. Affinity-selected phages were eluted using a low-pH solution (3 mg/mL BSA and 0.1 M glycine, pH 2.2) and immediately neutralized with 1 M Tris-HCl, pH 9.1. For phage amplification, *E. coli* DH5αF⁺ were infected with the affinity-selected phages and grown overnight at 37 °C in liquid Terrific Broth (TB) and phosphate buffer (PB) medium containing 20 μg/mL tetracycline. The infected bacteria were then centrifuged, and the supernatant containing phages was

incubated with 0.4 volumes of PEG-NaCl solution for 2 ho at 4 °C. Following incubation, the mixture was centrifuged, the supernatant discarded, and the affinity-selected phages were resuspended in TBS×1. Each well underwent three rounds of Biopanning, with amplification steps between rounds.

Dot blot analysis was performed to confirm the binding of mAb 7M1162 to affinity-selected phages. *E. coli* DH5αF⁺ were infected with affinity-selected phages from the third round of biopanning and grown overnight at 37 °C on Luria broth (LB) plates containing 20 μg/mL tetracycline (Holland Moran). Single bacterial colonies were individually picked and cultured overnight at 37 °C in TB + PB medium containing 20 μg/mL tetracycline. The cultures were then centrifuged, and the supernatant containing phages was incubated with 0.4 volumes of PEG-NaCl solution for 2 h at 4 °C. After incubation, the mixture was centrifuged, the supernatant discarded, and affinity-selected phages were resuspended in TBS×1. Single-colony-derived phages were transferred onto nitrocellulose membranes (Tamar Ltd.) using negative pressure. Membranes were blocked for 2 h at room temperature (RT) in a blocking solution containing TBS×1, 5% skim milk, and 0.05% Tween-20, followed by a single wash with TBS×1 and 0.05% Tween-20. Mab 7M1162 used for Biopanning was diluted to 2 μg/mL in blocking solution and incubated with the membranes for 1 h at RT. Following three washes with washing buffer, membranes were incubated with an anti-human IgG HRP-conjugated secondary antibody (1:5000 dilution in blocking solution) for 45 min at RT. After five additional washes, ECL substrate was added, and Western blot images were acquired. Dot blot-positive phages were regrown, isolated, and revalidated.

Plasmids containing the DNA of validated phages were isolated and sent for Sanger sequencing to determine the peptide insert on the pVIII protein. Unique peptide sequences were used for epitope prediction based on the VACV H3 structure PDB:5EJ0[88] using the PepSurf[55] software.

## Statistical Analysis

Statistical analysis was carried out using the GraphPad Prism software version 9.4.1. Analysis of Flow Cytometry data was carried out using FlowJo software version 10.8. Fluorescent cell images were analyzed using the IncuCyte live-cell analysis system for VACV and Harmony software for MPXV. Protein 3D structures were visualized using Pymol software version 3.1.3.1. Statistical details of each experiment can be found in the figure legend associated with each figure.

## Ethics statement

Peripheral whole blood samples were collected at 1–2 months and 9–10 months post-Mpox infection or primary vaccination, as detailed in Supplemental Table 1. The samples used for most analyses in this study are listed in Supplemental Table 1, except for those used in Supplemental Fig. 1, which were obtained from a cohort previously described[18]. All Mpox-convalescent and vaccinated donors provided written informed consent before sample collection at each time point. All human participants in this study were male individuals, reflecting the epidemiology of the MPXV sub-clade IIb outbreak in Israel during 2022, which disproportionately affected male individuals. Sex was determined from medical records at the time of diagnosis. All Mpox-convalescent donors presented with mild disease, consistent with the clinical course of the outbreak in Israel. Gender identity was not separately assessed, as it was not relevant to the study objectives. Participants did not receive any financial or material compensation for their participation in this study. The Tel Aviv University Institutional Review Board (IRB) approved all studies involving patient enrollment, sample collection, and clinical follow-up (protocol number 0005243–1; Helsinki approval number 0384-22-TLV). In vivo studies complied with the Tel Aviv University Institutional Animal Care and Use Committee (permit numbers TAU-MD-IL-2407-149–4, TAU-MD-IL-2411-181–5, and TAU-MD-IL-2502-104–5).

## Reporting summary

Further information on research design is available in the Nature Portfolio Reporting Summary linked to this article.

## Data availability

The X-ray crystallography data generated in this study have been deposited in the Protein Data Bank (PDB) under accession codes 9SHD for the A26−10M2146 complex and 9QT3 for the A26−8M2110 complex. All other data supporting the findings of this study are available within the paper and its Supplementary Information files. The Source Data file provided with this paper includes all raw data generated and analyzed in this manuscript. Source data are provided with this paper.

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

## Acknowledgements

We thank Bernard Moss for providing us with the VACV strains: VACV-WR-vFire, VACV-IHDJ-vFire and VACV-Copenhagen-vFire. We thank Ohad Mazor, Tomer Israely and Hadas Tamir from the Israel Institute for Biological Research for their support with reagents and advice. We thank TAU international student, Yonathan David Shternberg, for assisting with the project. The authors acknowledge the scientific and technical assistance at the Faculty of Medical & Health Sciences Research Infrastructure Core Facilities (RICF) of The Gray Faculty of Medical and Health Sciences, particularly Dr. Irena Shur and Dr. Daria Makarovsky from the RICF for their professional and valuable help with Flow Cytometry resources and Single B Cell Sorting. We thank Dr. Vered Holdengrerber for her assistance with TEM measurements. We thank the staff of the Institut Pasteur crystallography facility and the PX1 and PX2 beamlines at synchrotron SOLEIL (St. Aubin, France). We thank Eli Gelman, Tel Aviv University's Executive Council, and Ariel Porat, the President of Tel Aviv University for their support at the beginning of this project. We thank Israel Science Foundation (ISF) grants [3136/22] and [638/23] to NTF; Binational Science Foundation (BSF) [01031771] to NTF; BMGF INV-058519 to NTF. RY was supported by a PhD Scholarship from the Tel Aviv University Center for Combatting Pandemics. KP's research is supported in part by a fellowship from the Edmond J. Safra Center for Bioinformatics at Tel Aviv University. NR thanks the Sagol Center for Regenerative Medicine for financial support. The OS lab is funded by Institut Pasteur, Fondation pour la Recherche Médicale (FRM), ANRS-MIE, the Vaccine Research Institute (VRI) (ANR-10-LABX-77), Labex IBEID (ANR-10-LABX-62-IBEID), the HERA projects DURABLE (grant 101102733) and LEAPS. The P.G.-C. lab is funded by the Institut Pasteur and the National French Research Agency (ANR; ANR-22-CE11-0003 and ANR-24-CE15-6625). Lastly, we thank all the human blood donors who participated in this study.

## Author contributions

N.T.F. and R.Y. conceived the project, planned and supervised the experiments and data analyses, and wrote the manuscript together with P.G.C. and O.S. N.F. collected human whole blood samples with assistance from DH and ES, and R.Y. processed the samples. RY and NBS performed antigen production, single B cell sorting, monoclonal antibody cloning, and binding characterization experiments. R.Y. and K.R. carried out VACV propagation with assistance from M.R.A., S.S., and T.K., under the supervision of O.K. In vitro VACV and MPXV assays were performed by R.Y., K.R., M.H., F.G.B., F.P., and J.P. Z.F. provided critical feedback on the complement assays. In vivo experiments were conducted by R.Y., L.A., and K.R. L.B. performed all structural analyses, data collection, and figure preparation, with assistance from P.G.C. R.Y., G.O., and K.P. conducted Biopanning experiments and in silico modeling. R.Y. and N.R. performed the TEM experiments under the supervision of LAA.

## Competing interests

The authors declare no competing interests.
