## [Transparent Peer Review file · Nature Communications]

Potent Neutralization by Antibodies Targeting the MPXV A28 Protein

Corresponding Author: Professor Natalia T Freund

Editorial Note: Parts of this peer review file have been redacted as indicated to avoid any copy right infringement.

Version 0:

Reviewer comments:

Reviewer #1

(Remarks to the Author)

Review for "Potent Neutralization by Antibodies Targeting the Mpox A28 Protein" by Yefet et al. This manuscript presents a tour de force study highlighting compelling and novel data about a critical and understudied surface determinant, A28, in the protective immune response against Mpox, targeted in most convalescent individuals. Specifically, building on previous data related to the protection afforded by commonly targeted antigens, including A35, H3, etc.. Yefet et al. note the presence of a robust and immunodominant response to A28. Antibodies were cloned against A35, H3, and A28, revealing convergent neutralizing antibody evolution across donors. Moreover, specific analysis of A28-specific monoclonal antibodies revealed that A28 responses were robustly neutralizing, often outperforming other specificities, all in a complement dependent manner. These antibodies were mostly cross-reactive to VACV, an ortholog of Mpox. Germline reversion highlighted how affinity maturation broadened the cross-neutralizing potency of these antibodies. Moreover, combinatorial studies demonstrated the synergistic effects of A28 cocktails with other nearly all tested mAbs. Structural analysis was performed to map the specificities of these novel A28 antibodies, highlighting targeting of a common site. While the antibodies conferred partial protection in vivo, vaccination with this target revealed robust and complete protection against the virus. These data argue that a polyclonal response to A28 may be required to drive protection.

Collectively, this study opens the aperture in the Mpox field to think of additional protective specificities that are critical for next generation therapeutic and vaccine development. The authors here perform a deep and rigorous study profiling these A28 responses in the context of the previously highly studied responses. The study is done with remarkable rigor and provide critical clues and insights on the potential mechanism of protection of these novel antibody specificities, helping the field build a better understanding of the correlates of protection against Mpox. This is a very well-done study that adds truly novel information to the field and sets a new standard for antibody analyses. A few minor comments are listed below:

1. While the A28 vaccine induced antibodies were fully protective against lethal VACV challenge, disease was not attenuated in the animals. Some more discussion would be helpful to try to contextualize. Do these antibodies bind to a broader landscape of epitopes on the antigen compared to the monoclonal antibodies? Do they have additional effector functions compared to the monoclonal? Do they perhaps have higher affinity? Or is the size of the immune complex critical for driving enhanced complement mediated protection. This result is quite compelling and an important step in our understanding of the correlates of protection against Mpox. Some additional discussion on these different possibilities could also be helpful for readers.
2. Although sequence differences between VACV A26 and Mpox A28 were noted, it is unclear whether the ¼ A28 antibodies that didn't neutralize may hit a distinct epitope. This may provide important insights for vaccine design if A28 would be included.
3. It is unclear how often A28 responses are observed in vaccinated individuals. A little more detail on immunodominance profiles across convalescent and vaccinated individuals would be highly informative in comparison to other specificities. Additionally, are these responses amplified in individuals that experience more severe disease?
4. Some discussion could be added on why A28 would be useful as an additional antigen determinant in current multi-component vaccines given that they already provide complete protection in primates. What additional value would A28 add.
5. For the first MPOX call out in the abstract- it would be helpful to spell it out.

Reviewer #2

(Remarks to the Author)

This is an interesting manuscript that mainly focuses on a mature virus (MV) surface protein called A26 (using vaccinia

nomenclature also known as A28 mpox nomenclature, also known as OPG 153). This MV surface protein is not membrane bound but is on the virion surface through an interaction with membrane bound A27 protein. Interestingly, A26 is "non-essential" for growth, replication and spread in cell culture and the deletion mutant in virulent vaccinia virus strain WR has actually increased virulence than the parental virus. It has not previously been investigated as a target of neutralizing antibodies.

The authors isolate human monoclonal antibodies to an EV specific protein A33 and two MV-specific proteins H3 and A26 (vaccinia nomenclature) from mpox-infected patients. While data is included related to monoclonals to A33 and H3, the main focus of the paper are the antibodies to A26. Actually, the A33 data is a bit confusing given that the neutralization experiments appear to be using MV. That makes it hard to explain the synergistic effect of anti-A33 MAb in some assays.

The strength of this manuscript is the plentiful data and the multiple different assays and approaches. Authors also test the protection of mice after vaccination using purified A26 protein in adjuvant.

Major comments:

1. Given the complexity of the data and the complex names of monoclonal antibodies, for readability, would suggest that when a MAb is mentioned in the text, the protein target is included.
2. The paper does not adequately distinguish between MV and EV when discussing work with virus and neutralization assays, etc. Also, need to better state when sucrose gradient purified virus (MV) is used and when virus from infected cell lysate that had undergone freeze-thaws (essentially MV) is used.
3. Given data presented, should title of the paper be: Potent Complement-dependent Neutralization by Antibodies Targeting the Mpox A28 Protein

Minor comments:

1. Source of normal human sera/complement is not described. Could it have been from people who had previously received smallpox vaccination?
2. Authors do experiments with human and guinea pig complement, but they should better highlight the species being used. Perhaps including that information in figures (not just legend) would help.
3. Line 43. the emergence of a new Mpox clade (clade IIb). Should it be sub-clade (clade IIb)?
4. Line 322/323. 1.5% complement likely not high enough to see complement mediated cellular cytotoxicity.
5. Line 324/325. Reference 55 is complement-mediated neutralization of EV. Authors are mainly looking at complement-mediated neutralization of MV, where coating of virion with complement could block infection .
6. Line 370. Title of section is misleading since first paragraph describes the exact opposite. Better section header might be "Active vaccination with A28 protein protects With that said, have authors considered that passive transfer with human antibody did not adequately activate mouse complement and/or mouse FcR binding as the explanation for lack of efficacy in the mouse model?
7. Line. 427. While reference 56 describes the importance of A26 during a viral infection, some of the functions are during intracellular virion maturation which occurs late in infection. Are authors suggesting that some of the MAb could act intracellularly?
8. Line 624 and 670. What table 1 and vaccinated subjects are they referring to? Supplemental Table 1 is just infected subjects. Are they referring to sera used in Supplemental figure 1A? Were there samples from JYNNEOS vaccinated subjects in this work?
9. Line 774. Not clear where inactivated virus was used for ELISA. Would add info to figure legend that inactivated IHDJ was used.
10. Line 825. Section should make it clearer what form of virus was being neutralized. Suspect it was MV.
11. Line 829. Do not understand if neutralization was done in the presence of 6% or 1.5% NHS? I believe the standard way of determining % complement would be the amount that is incubated with virus and antibody (prior to adding to cell culture to determine the remaining infectious titer)
12. Line 843. Similarly, do not understand if mpox neutralization was done in the presence of 40% or 10% guinea pig complement?
13. Line 908. What was the source of IHDJ virions were used (e.g., mainly MV or could this be EV)?
14. Figure 1B. Why such a large size difference between the expressed mpox and VACV A33 protein?
15. Figure 1H, table. Why does first column show 8M2110 and 10M2146 not competing, but second column show they do compete? Later crystal structures show binding sites are close and authors state this explains the competitive binding observed
16. Figure 1 colored lines and colored data points on graphs. Shades of each color used is difficult to distinguish. Can the MAbs that moves forward be the darkest shade and perhaps their names bolded in the legend? Seems like anti-A26 MAbs 8M2110 and 10M2146 and anti-H3 7M1162 are the ones used in multiple assays.
17. Figure 2C table. Are values in the table based on graph in Fig 2B or 2C or both? Reason I ask is anti-H3 MAb 5M1111 only looks to neutralize with 10% guinea pig complement (and not active with 1.5% human complement).
18. Figure 3. Legend for panel F is missing.
19. Figure 5F & 5G. Were there any controls of virions with complement but no antibody or a control non-binding antibody?
20. Supplemental figure 1. Title of figure should not include IHDJ.
21. Supplemental Figure 3. Might be nice to include sequence for Copenhagen homolog to help explain non-binding of

MAbs described in this paper.

Reviewer #3

(Remarks to the Author)

In this manuscript, the authors isolated antibodies to the Mpx antigens A28, A35, and H3 from convalescent donors. They determined that anti-A28 antibodies were able to neutralize Mpx and VACV through complement-dependent mechanisms and showed the crystal structures of two anti-A28 antibodies to the VACV homologue A26. The binding sites were distinct but highly conserved between VACV and Mpx. They also show that vaccination with Mpx A28 produced antigen-specific B-cells and high neutralization responses to VACV infection in mice.

Overall, I found the biological data to be well done and convincing. The findings are also timely and valuable to the field. However, there are some issues with the structural data that need clarification and refinement prior to publication.

Major issues:

- 1) Some reported statistics in Table S3 are below acceptable standards. For the A26 + Fab 10M2146 data set the authors report an $\langle l/s \rangle$ of 0.5 in the highest resolution shell. This can be acceptable if the CC1/2 is sufficient but the reported value is 0.21 in the highest resolution shell is really pushing it to the extreme limit suggesting the resolution is lower than reported. There are also numerous discrepancies between the text, Table S3 and the validation reports. Some differences are fine as the values in the validation report and values from phenix.refine or MolProbity are calculated differently but should be within reason. In the text the resolution for A26 + Fab 10M2146 is reported to be 1.9A but stated as 1.80A elsewhere. The validation report for A26 + Fab 10M2146 reports a $\langle l/s \rangle$ of 1.27 with 0% Ramachandran outliers but Table S3 has these values of 0.5 and 2.34%, respectively. If the $\langle l/s \rangle$ is actually 1.27 with a CC1/2 above 0.3 this resolution would be acceptable.
- 2) Starting with the paragraph on line 118 and Fig1 K-M discuss the role of antibody maturation and effects of germline revision on binding to Mpx or VACV. It would be useful to extend this analysis to the structures of 8M2110 and 10M2146. Do the somatically mutated residues explain why 8M2110 loses binding to Mpx A28 but not VACV A26 and why 10M2146 binding is unaffected? The inclusion of a sequence alignment of the mature and germline mAbs highlighting interacting residues would be helpful.
- 3) With the major focus of the manuscript being on Mpx, can the authors comment on why the structure was not determined with Mpx A28. Was it tried and didn't work or was VACV A26 chosen initially and why was that used?
- 4) The inclusion of the epitope prediction and modelling of H3 and 7M1162 is confusing, not necessarily informative and has several issues. In the text the cluster are referred to as "clusters (1-3)" but in the figure they are clusters A-C. The methods say that the AlphaFold3 model was generated with Mpx H3 and 7M1162 but the VACV H3 model is shown in FigS4. Was AF3 used to predict the binding to VACV H3 and, if so, how does it compare to that of Mpx H3. How AF3 was used needs to be much more specific. Was the webserver used or local version? If a local version was used, how many models were generated and how many seeds were used? How was the binding model shown in FigS4 chosen, like what metrics were used to decide on this one? Was the model done with full Fab or just variable region of 7M1162?

Minor issues:

- 1) Throughout the figures, the slight differences in green color for the A28 binding antibodies is very hard to distinguish between.
- 2) The significant figures in Table S3 should be consistent. For example, the unit cell dimensions for one data set are reported to 1 decimal place but 3 in the other.
- 3) It is reported that TLS refinement was used. How many TLS groups were used per chain?
- 4) The methods mention the use of an anti-Fab VHH in the complex but none of the figures in the main text or supplemental show this anti-Fab VHH in the structural model. While not necessarily relevant to the conclusions of the paper, a figure showing the entire complex or contents of the ASU should be included.
- 5) In the methods it says the initial phases were determined with a published the A26 model. Were the Fab or anti-Fab VHH included in phasing? If not, how were the Fab and anti-Fab VHH models initially built and placed?
- 6) This manuscript was likely submitted before the PDB announcement of transitioning to Extended PDB IDs (<https://www.rcsb.org/news/feature/6875133f3b59581b68019794>). I would recommend updating all references to PDB ID to this extended format.

Version 1:

Reviewer comments:

Reviewer #1

(Remarks to the Author)

The authors have adequately addressed the reviewers comments. The paper remains strong and of broad interest to both MPXV vaccinologists and the broader virology community.

Reviewer #2

(Remarks to the Author)

Authors provide an excellent and detailed response to reviewers' comments and have appropriately adjusted text in the manuscript.

I appreciated the authors' explanation of how they report the concentration of complement used in their neutralization assays. However the reference they cite in the methods (ref 46; line 875/876) uses the complement concentration during incubation with virus prior to adding to cells to measure the amount of MV neutralized. I believe that would be the more standard way to report the % complement included during a virion neutralization assay. Note that there is approximately 2-log decrease in MV titer within the first 30 mins of incubation of vaccinia virus with antibody and complement (PMID 1731333). Also note that ref 75 may not be a correct reference in the methods section.

Minor comments to consider to enhance current text and focus mostly on the abstract.

1. Line 28 (and 68) Curious about describing A26 as a virulence factor when deletion of the gene actually increased virulence when compared to the parental virus in mouse model. Nuanced description of A26 functions in the discussion is more accurate. Authors should consider A28 as a key or important virion protein.
2. Line 33. Abstract says, MAb attenuated disease in infected mice, but then later (line 80 or section starting on line 391), the MAb treated mice yielded only modest, non-significant clinical benefit in vivo. Perhaps better said, Passive transfer of 8M2110 had a trend toward attenuating disease in infected mice.
3. Line 35 (and 408) Would not characterize the protection provided by active immunization as complete given significant weight loss. Would just say protection.
4. Line 58/59. Would include OPG names for A29, E8, H3, M1, A35 and B6
5. Line 198/199. Based on new Figure S3, looks like Copenhagen A26 had a frameshift mutation that results in altered C-terminus amino acid sequence as well as a truncation.
6. Line 688. What experiments used sucrose purified MV particles? Was it just virions used for electron microscopy. So instead of saying experiments, would say here what experiments used lysate that were pelleted through a sucrose cushion.
7. Authors' response letter have a few revised Figure panels that has panels that are not in the manuscript.

Reviewer #3

(Remarks to the Author)

The authors have adequately addressed all of my comments and concerns. This is a very well done study.

We thank the reviewers for their thoughtful assessment of our manuscript, “**Potent Neutralization by Antibodies Targeting the Mpox A28 Protein**” for publication in *Nature Communications*. We are encouraged that they found the study to be novel, therapeutically relevant, and rigorous, and we appreciate their recognition of the high quality of the data. We have carefully considered and addressed all comments in the revised manuscript and in a point-by-point response below.

Color code:

Reviewer’s text

Authors’ response

Changes in the text

Reviewer #1 (Remarks to the Author):

Review for “Potent Neutralization by Antibodies Targeting the Mpox A28 Protein” by Yefet et al. This manuscript presents a tour de force study highlighting compelling and novel data about a critical and understudied surface determinant, A28, in the protective immune response against Mpox, targeted in most convalescent individuals. Specifically, building on previous data related to the protection afforded by commonly targeted antigens, including A35, H3, etc.. Yefet et al. note the presence of a robust and immunodominant response to A28. Antibodies were cloned against A35, H3, and A28, revealing convergent neutralizing antibody evolution across donors. Moreover, specific analysis of A28-specific monoclonal antibodies revealed that A28 responses were robustly neutralizing, often outperforming other specificities, all in a complement dependent manner. These antibodies were mostly cross-reactive to VACV, an ortholog of Mpox. Germline reversion highlighted how affinity maturation broadened the cross-neutralizing potency of these antibodies. Moreover, combinatorial studies demonstrated the synergistic effects of A28 cocktails with other nearly all tested mAbs. Structural analysis was performed to map the specificities of these novel A28 antibodies, highlighting targeting of a common site. While the antibodies conferred partial protection in vivo, vaccination with this target revealed robust and complete protection against the virus. These data argue that a polyclonal response to A28 may be required to drive protection. Collectively, this study opens the aperture in the Mpox field to think of additional protective specificities that are critical for next generation therapeutic and vaccine development. The authors here perform a deep and rigorous study profiling these A28 responses in the context of the previously highly studied responses. The study is done with remarkable rigor and provide critical clues and insights on the potential mechanism of protection of these novel antibody specificities, helping the field build a better understanding of the correlates of protection against Mpox. This is a very well-done study that adds truly novel information to the field and sets a new standard for antibody analyses. A few minor comments are listed below:

We sincerely thank the Reviewer for their time to read our study and for the positive and encouraging evaluation of our manuscript. We are grateful for their recognition of the depth, rigor, and novelty of our study, and for highlighting that our work adds novel information to the field. As outlined below, we have carefully addressed the all the points raised by the Reviewer.

1. While the A28 vaccine induced antibodies were fully protective against lethal VACV challenge, disease

was not attenuated in the animals. Some more discussion would be helpful to try to contextualize. Do these antibodies bind to a broader landscape of epitopes on the antigen compared to the monoclonal antibodies? Do they have additional effector functions compared to the monoclonal? Do they perhaps have higher affinity? Or is the size of the immune complex critical for driving enhanced complement mediated protection. This result is quite compelling and an important step in our understanding of the correlates of protection against Mpox. Some additional discussion on these different possibilities could also be helpful for readers.

We thank the Reviewer for this important note. As correctly pointed out, A28 vaccine–induced antibodies were fully protective against lethal VACV challenge but did not attenuate disease, as reflected by the observed weight loss in the immunized animals. While this was a proof-of-concept study demonstrating the relevance of A28 as an immunogen, the lack of disease attenuation can be explained by the fact that A28 is expressed only on mature virions (MV); therefore, antibodies elicited by immunization are unlikely to target extracellular virions (EV), allowing viral dissemination. In this context, a combination vaccine including both A28 and EV antigens is expected to broaden coverage and reduce potential viral spread.

To explore whether vaccination-induced antibodies recognize a broader range of epitopes than isolated monoclonal antibodies, we performed two competition ELISA analyses using sera from A28-immunized mice and our A28-specific mAbs.

*In the first analysis, we tested whether vaccination elicited antibodies that bind in the same site as our A28-specific mAbs. Sera from vaccinated mice competed with isolated mAbs (8M2110, 10M1135, 10M2146, 10M2154), showing that the epitopes targeted by A28-specific mAbs isolated from convalescent donors are also recognized by the polyclonal response elicited following A28 immunization. Competition across multiple serum dilutions further demonstrated the potency of the vaccine-induced antibodies (**Revision Figure 1A**). In the second analysis, we tested whether vaccination also elicited antibodies against additional epitopes on A28. For this, A28 was first blocked with mAbs (8M2110, 10M2146, or both), then mouse sera was tested for binding. Sera from two vaccinated mice bound Mpox A28 even after prior binding by mAbs indicating that vaccination induces a broader polyclonal response that targets epitopes beyond those recognized by isolated mAbs (**Revision Figure 1B**). As the Reviewer correctly noted, Fc-effector functions or properties of the immunogen may also contribute to the protection observed following A28 immunization. We plan to address these possibilities in future studies, focusing on the roles of Fc-mediated activity, immunogen design, and polyclonal antibody breadth.*

A section addressing this point has now been added to the Discussion section (lines 497-501):

“The superior protection observed with vaccination compared to mAb treatment may stem from a broader polyclonal response that targets additional epitopes beyond those recognized by the mAbs. It may also reflect properties of the immunogen or enhanced Fc-mediated functions acting in synergy with the immune system. These mechanisms require further investigation but support the relevance of A28 as a promising subunit for next-generation Poxvirus vaccines.”

Revision Figure 1. Mouse sera recognizes both dominant and subdominant epitopes on A28. (A) Blocking anti-A28 mAbs binding to A28 by vaccinated mice sera. A28-coated wells were incubated with each mAb (8M2110, 10M1135, 10M2146, or 10M2154) at a fixed concentration of 0.1 $\mu\text{g}/\text{mL}$ in the presence of five 4-fold serial dilutions of sera from A28-vaccinated mice, starting at 1:40. Binding was detected using an anti-human HRP-conjugated secondary antibody, and absorbance was measured at OD_{650} . The mean response of sera from eight A28-vaccinated mice is shown in bold, and shaded areas represent the standard deviation across individual serum samples. **(B)** Blocking of vaccinated mice sera binding to A28 by anti-A28 mAbs. Binding of sera from A28-immunized mice (Sera 5.3 and Sera 6.2) was measured in the presence of a 4-fold serial dilution of individual or combined anti-A28 mAbs (8M2110, 10M2146, or 8M2110+10M2146) and an irrelevant control mAb (MGO53), starting from 32 $\mu\text{g}/\text{mL}$. Binding was detected using an anti-mouse HRP-conjugated secondary antibody, and absorbance was measured at OD_{650} .

2. Although sequence differences between VACV A26 and Mpx A28 were noted, it is unclear whether the $\frac{1}{4}$ A28 antibodies that didn't neutralize may hit a distinct epitope. This may provide important insights for vaccine design if A28 would be included.

We thank the Reviewer for this important point. 10M1135 does not compete with A28 mAbs 8M2110 or 10M2146 (Figure 1H). Moreover, while 10M1135 binds Mpx A28 efficiently, it exhibits only weak binding to its VACV homolog (Figure 1G), and accordingly demonstrates a weak anti-VACV neutralization. To further investigate this antibody on a functional level, we repeated the Mpx neutralization assays while this time including an additional Mpx sub-clade, thus performing the neutralization assays against both Mpx clades 1b and 2b. In the presence of 1% NHS 10M1135 can in fact neutralize both Mpx sub-clades, albeit less potently than 8M2110 or 10M2146, while showing no neutralizing activity against any VACV strain. Collectively, these findings suggest, as correctly pointed out by the Reviewer, that 10M1135 recognizes a distinct epitope on Mpx A28 that is absent, or poorly conserved, in VACV A26. The new data was now added to the new version of Figure 2 within the revised manuscript.

The text in the Results section was revised accordingly (lines 182-210):

"We next evaluated the neutralizing capacity of our mAbs against Mpx (sub-clades 1b and 1b), as well as against three VACV strains, WR, Copenhagen, and IHDJ using virus preparations enriched for MV. Several mAbs demonstrated neutralizing activity against Mpx or VACV, which in all cases was complement-dependent, except for the anti-A35 mAb 4M2131 that demonstrated weak but detectable

neutralization against VACV WR without complement (Figure 2C). Complement-dependent neutralization is consistent with previous reports of human antibodies elicited following infection^{18,22,26,46}. Among the tested mAbs, those targeting A28 exhibited the most potent neutralization, though their efficacy varied between Mpx sub-clades and VACV strains. For the anti-A28 mAbs: the clonal pair 10M2146 and 10M2154 neutralized Mpx sub-clade Ib with IC₅₀ values of 0.01 µg/mL for both mAbs, and sub-clade IIb with IC₅₀ values of 81.1 µg/mL and 0.98 µg/mL, respectively. MAb 8M2110 neutralized Mpx sub-clade Ib with an IC₅₀ of 0.01 µg/mL and sub-clade IIb with an IC₅₀ of 3.81 µg/mL. Meanwhile, the mAb 10M1135 neutralized Mpx sub-clade Ib with an IC₅₀ of 0.43 µg/mL and sub-clade IIb with an IC₅₀ of 3.28 µg/mL (Figures 2A-2B and 2E). The mAbs also neutralized VACV: 10M2146 and 10M2154 neutralized the IHDJ strain with IC₅₀ values of 2.61 µg/mL and 72.58 µg/mL, respectively; 8M2110 neutralized WR and IHDJ with IC₅₀ values of 0.06 µg/mL and 0.16 µg/mL, respectively; whereas 10M1135 showed no detectable neutralization against any VACV strain (Figures 2C–2E). VACV Copenhagen was resistant to all tested anti-A28 mAbs, likely a result of the absence of the A26 antigen on the viral surface due to a truncation of its C-terminal region (Figure S3)⁴⁷⁻⁴⁹. The ability of anti-A28 mAbs to neutralize both the VACV WR and IHDJ strains highlights the potential of mAbs directed against A28 to exhibit broad pan-Poxvirus neutralization. To our knowledge, this is the first report to describe human Mpx-neutralizing mAbs targeting the A28 protein or its homologs.

Among the anti-H3 mAbs, 7M1162 was the only one to neutralize all three tested VACV strains, with IC₅₀ values of 1.18 µg/mL, 8.85 µg/mL, and 16.93 µg/mL against WR, Copenhagen, and IHDJ, respectively, but it did not neutralize either Mpx sub-clade. In contrast, the Mpx-specific mAb 5M1111 (anti-H3) neutralized Mpx with IC₅₀ values of 32.71 µg/mL and 7.61 µg/mL for sub-clades Ib and IIb, respectively, but had no activity against VACV (Figure 2). As might be expected, in this assay, mAbs targeting A35 exhibited minimal detectable neutralization activity, with measurable IC₅₀ values only for mAbs 4M1224 and 4M2131 against Mpx sub-clade Ib (80.57 µg/mL and 29.11 µg/mL, respectively- Figure 2A-2B). This is likely due to the assay's primary focus on MV as opposed to EV."

3. It is unclear how often A28 responses are observed in vaccinated individuals. A little more detail on immunodominance profiles across convalescent and vaccinated individuals would be highly informative in comparison to other specificities. Additionally, are these responses amplified in individuals that experience more severe disease?

We thank the Reviewer for this question. Data regarding the serological response to Mpx A28 are shown in Figure S1, while serologic activity against H3, M1 and A35 of this exact cohort was reported in our previous publication (Yefet et al., iScience 2023). In this study, we compared the serological responses of Mpx-convalescent individuals to MVA-vaccinated donors, historical Smallpox vaccine recipients (vaccinated with VACV, <1977), and naïve controls (>1977). Analysis across the cohort indicates that A28 responses are consistently observed in convalescent individuals, though their magnitudes vary (Revision Figure 2). Responses in vaccinated individuals are generally lower compared to convalescent donors, consistent with limited antigen exposure during vaccination, as MVA (like the VACV Copenhagen strain) does not express the VACV homolog on its surface. The magnitude and frequency of A28-specific responses in convalescent donors are similar to those observed for A35, a known immunodominant target, highlighting that A28 is likewise an immunodominant antigen and underscoring its relevance in both natural infection and vaccination contexts. We did not include this data in the main manuscript since part of the dataset has already been published.

[redacted]

As to the Reviewer's second question, regarding diseases severity, due to the relatively mild disease course in all the Mpox-convalescent donors (who all were early treated and followed), a correlation between the A28-specific response and disease severity could not be established.

A section addressing this point has now been added to the Ethics statement section (lines 655-657):

"All Mpox-convalescent donors presented with mild disease, consistent with the clinical course of the outbreak in Israel."

4. Some discussion could be added on why A28 would be useful as an additional antigen determinant in current multi-component vaccines given that they already provide complete protection in primates. What additional value would A28 add.

We thank the Reviewer for the opportunity to highlight the potential of A28. A section addressing this point has now been added to the Discussion (lines 491-495):

"Including A28 in such formulations could enhance the resilience of the immune response, particularly given the extensive APOBEC3-driven mutational changes observed in Mpox, which could lead to antigenic escape in other vaccine components^{72,73}. Moreover, our findings indicate that antibodies against A28 are largely germline-encoded, requiring minimal affinity maturation, suggesting they can be readily induced by vaccination."

5. For the first MPOX call out in the abstract- it would be helpful to spell it out.

Thank you. This point has now been addressed in the Abstract and Introduction sections.

Abstract (lines 26-27):

“Monkeypox virus (Mpox) is the most pathogenic Poxvirus in circulation, yet key viral antigens remain immunologically unexplored.”

Introduction (line 39-40):

“Monkeypox virus (Mpox) is a member of the Poxvirus family, an infamous viral lineage which has been a major source of human morbidity throughout history^{1,2}”

Reviewer #2 (Remarks to the Author):

This is an interesting manuscript that mainly focuses on a mature virus (MV) surface protein called A26 (using vaccinia nomenclature also known as A28 mpox nomenclature, also known as OPG 153). This MV surface protein is not membrane bound but is on the virion surface through an interaction with membrane bound A27 protein. Interestingly, A26 is “non-essential” for growth, replication and spread in cell culture and the deletion mutant in virulent vaccinia virus strain WR has actually increased virulence than the parental virus. It has not previously been investigated as a target of neutralizing antibodies. The authors isolate human monoclonal antibodies to an EV specific protein A33 and two MV-specific proteins H3 and A26 (vaccinia nomenclature) from mpox-infected patients. While data is included related to monoclonals to A33 and H3, the main focus of the paper are the antibodies to A26. Actually, the A33 data is a bit confusing given that the neutralization experiments appear to be using MV. That makes it hard to explain the synergistic effect of anti-A33 MAbs in some assays. The strength of this manuscript is the plentiful data and the multiple different assays and approaches. Authors also test the protection of mice after vaccination using purified A26 protein in adjuvant.

We thank the reviewer for their time and effort to thoroughly assess our study, and for the meticulous evaluation and constructive feedback on our manuscript. We appreciate their recognition of the strength of our study, particularly the breadth of experimental approaches and the inclusion of both monoclonal antibody characterization and vaccination experiments.

As the reviewer correctly noted, the neutralization assays in this study (with either Mpox or VACV) were performed using viral particles obtained from lysed infected cells. This approach produces preparations highly enriched for MV particles. We acknowledge that optimal EV neutralization assays are typically performed using cell culture supernatants from IHDJ-infected cells, either in comet inhibition assays or PRNT, in combination with an MV-neutralizing mAb to remove MV background. The minimal neutralization observed for A35-targeting mAbs (primarily 4M2131) and their synergistic effect with MV-targeting mAbs could be explained by the small proportion of EV particles still present in the viral preparations. Alternatively, we cannot rule out post-infection neutralization of EV particles released from infected cells, which might account for the inhibitory effect observed with some A35 mAbs.

A section addressing this point has now been added to the Limitations of study section (lines 512-515):

“Moreover, most of the neutralization assays performed in this study were MV-focused, which may underestimate the contribution of anti-A35 mAbs targeting EV particles to overall neutralization. These anti-A35 mAbs will be investigated in follow up studies. “

Major comments:

1. Given the complexity of the data and the complex names of monoclonal antibodies, for readability, would suggest that when a MAb is mentioned in the text, the protein target is included.

Thank you. This point has now been addressed throughout the text.

2. The paper does not adequately distinguish between MV and EV when discussing work with virus and neutralization assays, etc. Also, need to better state when sucrose gradient purified virus (MV) is used and when virus from infected cell lysate that had undergone freeze-thaws (essentially MV) is used.

We thank the Reviewer for this suggestion. This point has now been clarified and addressed throughout the text in the Results, Methods and the Limitations of study sections.

Results (lines 182-183):

"We next evaluated the neutralizing capacity of our mAbs against Mpox (sub-clades Ib and IIb), as well as against three VACV strains, WR, Copenhagen, and IHDJ using virus preparations enriched for MV."

Methods (lines 693-695 and 920-921):

"In all assays using infectious Mpox or VACV, the viral particles used were MV-enriched preparations obtained by three freeze–thaw cycles, without further purification, unless otherwise specified."

"All viral preparations used in these assays were enriched for MV particles, as described above."

Limitations of study (lines 512-515):

"Moreover, most of the neutralization assays performed in this study were MV-focused, which may underestimate the contribution of anti-A35 mAbs targeting EV particles to overall neutralization. These anti-A35 mAbs will be investigated in follow up studies. "

3. Given data presented, should title of the paper be: Potent Complement-dependent Neutralization by Antibodies Targeting the Mpox A28 Protein

We thank the Reviewer for this suggestion. While the monoclonal antibodies described in our study neutralized exclusively in the presence of complement, the polyclonal sera of mice immunized with both Mpox A28 and VACV A26 demonstrated neutralization in the absence of complement and following heat inactivation. That suggests that A26/28 is a target for antibodies that can neutralize either with or without complements components. Therefore, after giving it much thought we respectfully disagree with the reviewer and would like to keep the original title of the manuscript.

Minor comments:

1. Source of normal human sera/complement is not described. Could it have been from people who had previously received smallpox vaccination?

*We thank the Reviewer for this question. The normal human sera (NHS) used in this study was obtained from an external vendor (Sigma-Aldrich Cat#H4522), as described in the Methods section. According to the vendor, the sera was sourced from male donors within the United States and tested negative for HBV, HCV, and HIV. The Smallpox or Mpox vaccination status of the donors was not disclosed. To confirm that mAb-mediated neutralization was not dependent on antibodies present in the human sera, we also tested the mAbs in the presence of guinea pig complement (GPC), used primarily for Mpox neutralization, and in mouse sera (**Revision Figure 3**- also described in **Minor comment 6**).*

2. Authors do experiments with human and guinea pig complement, but they should better highlight the species being used. Perhaps including that information in figures (not just legend) would help.

This point has now been addressed throughout the Figures.

3. Line 43. the emergence of a new Mpox clade (clade IIb). Should it be sub-clade (clade IIb)?

We agree with the Reviewer and accept the correction. This point has now been addressed throughout the text and figures.

4. Line 322/323. 1.5% complement likely not high enough to see complement mediated cellular cytotoxicity.

We thank the Reviewer for this important point. Indeed, we agree that complement-mediated cellular cytotoxicity requires concentrations higher than 1.5% NHS to effectively lyse infected cells, and that higher levels might have elicited detectable CDC. However, the purpose of this assay was not to evaluate CDC activity per se, but rather to elucidate the mechanism of antibody mediated neutralization in the presence of 1.5% NHS observed in our assays. Importantly, we demonstrated that the mAbs were able to neutralize VACV in the presence of 1.5% complement, thereby supporting that the mechanistic role of complement in viral neutralization under these experimental conditions is not CDC.

5. Line 324/325. Reference 55 is complement-mediated neutralization of EV. Authors are mainly looking at complement-mediated neutralization of MV, where coating of virion with complement could block infection.

Indeed, reference (Cohen et al., PLoS One 2011) reports complement-mediated virolysis only for the EV form of VACV and not for MV. Our intention in citing this reference was to indicate that such a phenomenon has previously been described in the context of VACV.

To clarify in the text that this reference pertains specifically to the EV form, the text is now updated (lines 342-344):

“We next investigated whether the mAbs promote complement-dependent virolysis, a phenomenon previously reported for the EV form of VACV⁵⁵.”

6. Line 370. Title of section is misleading since first paragraph describes the exact opposite. Better section header might be "Active vaccination with A28 protein protects With that said, have authors considered that passive transfer with human antibody did not adequately activate mouse complement and/or mouse FcR binding as the explanation for lack of efficacy in the mouse model?

We agree with the reviewer’s comment that the current title of this section may be misleading. The title has therefore been changed accordingly to better reflect the results presented.

A revised section header was now updated in the text (line 390):

“Active vaccination with A28 protein mediates protection from lethal viral challenge in vivo”

Regarding the second point raised by the Reviewer, as the reviewer correctly noted, we cloned the mAbs into a human IgG1 backbone for all experiments, including the mouse studies. The possibility that their lack of in vivo efficacy was due to insufficient reactivity with mouse complement was indeed considered. To address this, we obtained normal mouse serum (NMS) and tested the complement-dependent neutralizing activity of the anti-A28 mAb 8M2110 (human IgG1) against VACV WR, the same strain used in the in vivo experiments (Revision Figure 3A). The neutralization profiles were comparable when using either NMS or NHS, indicating that the human IgG1 mAbs employed in our mouse studies are capable of activating mouse complement effectively.

To further investigate the lack of efficacy in vivo, we re-cloned the variable domains of the A28 mAbs into a mouse IgG2a backbone and tested their protective efficacy in vivo against 1×10^5 PFU of VACV IHDJ. As expected with this lower infectious dose, overall mortality was reduced in both the anti-A28-treated group (8M2110) and the isotype control group (MGO53). Nevertheless, the anti-A28 IgG2a mAbs did not demonstrate efficacy better than isotype control antibody in this setting either, suggesting that Fc receptor engagement was not a limiting factor in explaining their lack of protection in vivo (Revision Figure 3B).

Revision Figure 3. Assessment of Fc-mediated functionality of anti-A28 mAbs in relation to the animal model. (A) Complement-dependent neutralization of VACV WR by anti-A28 mAb 8M2110 (human IgG1 backbone) in the presence of either 1.5% normal human serum (NHS) or normal mouse serum (NMS), shown across six consecutive 4-fold dilutions starting from 62.5 $\mu\text{g/ml}$. (B) In vivo protection study following recloning of 8M2110 (anti-A28) and the isotype control MGO53 into a mouse IgG2a backbone. Mice were challenged with 1×10^5 PFU of VACV IHDJ, and survival was monitored over 11 days.

7. Line. 427. While reference 56 describes the importance of A26 during a viral infection, some of the functions are during intracellular virion maturation which occurs late in infection. Are authors suggesting that some of the MAb could act intracellularly?

We thank the Reviewer for raising this interesting point. Our mechanistic data support the conclusion that the anti-H3 and anti-A28 mAbs act on the viral surface by recruiting complement, rather than functioning intracellularly. That said, we acknowledge that antibodies can, in principle, exert effects inside infected cells through Fc-dependent interactions with intracellular receptors such as TRIM21. While our study does not provide evidence for such a mechanism, we are not excluding this possibility (and are planning to examine this in a future follow-up study). Rather, we highlight that the dominant neutralization mechanism observed here is complement-mediated activity on the virion surface.

The cited study of (Holley et al., Journal of Virology 2021) was intended to underscore the diverse and important roles of A26, including its involvement in the transition from EV to MV production, its role in attachment via laminin, and its pH-dependent suppression of the entry fusion complex.

The relevant section was now updated in the Discussion to avoid misinterpretation (lines 446-453).

“Although A28 is not essential for infection, it intrinsically regulates the transition from EV production, associated with intra-host spread, to MV production, which facilitates inter-host transmission⁵⁹. In addition, A28 promotes viral dissemination by mediating attachment through laminin binding and by suppressing premature activation of the entry fusion complex under low-pH conditions. Together, these multifaceted functions position A28 as a key contributor to viral spread and inter-individual transmission. The strong antibody response elicited following Mpx infection or vaccination further underscores its potential as a promising antigenic target for antibody-based interventions.”

8. Line 624 and 670. What table 1 and vaccinated subjects are they referring to? Supplemental Table 1 is just infected subjects. Are they referring to sera used in Supplemental figure 1A? Were there samples from JYNNEOS vaccinated subjects in this work?

We thank the Reviewer for the opportunity to clarify this point. The discrepancy noted by the reviewer is valid. To clarify: in Supplemental Figure S1 we examined the polyclonal antibody response against Mpx A28 using samples from a previously described cohort (Yefet et al., iScience, 2023). This cohort included Mpx-convalescent individuals, MVA-vaccinated donors, historical Smallpox vaccine recipients (vaccinated with VACV before 1977), and naïve controls (born after 1977). From this cohort, only samples from Mpx-convalescent donors (collected either 1–2 months or 9–10 months post infection, as detailed in Table 1) were subsequently used for B cell isolation in the present study.

The relevant section was now updated in the Ethics statement to avoid confusion (lines 649-653).

“Peripheral whole blood samples were collected at 1–2 months and 9–10 months post-Mpx infection or primary vaccination, as detailed in Table 1. **The samples used for most analyses in this study are listed in Table 1, except for those used in Figure S1, which were obtained from a cohort previously described¹⁸.** All Mpx-convalescent and vaccinated donors provided written informed consent before sample collection at each time point.”

9. Line 774. Not clear where inactivated virus was used for ELISA. Would add info to figure legend that inactivated IHDJ was used.

The relevant section was now updated in the Supplemental figure 5D legend (lines 586-587).

“(D) Binding curves of anti-Mpox mAbs to β pL-inactivated VACV IHDJ as measured by ELISA of 4 consecutive 4-fold dilutions, starting from 100 μ g/mL.”

10. Line 825. Section should make it clearer what form of virus was being neutralized. Suspect it was MV.

The relevant section was now updated in the Methods section (lines 856-860).

“VACV strains: WR (VACV-vFIRE-WR), Copenhagen (VACV-vFIRE-Copenhagen), and IHDJ (VACV-vFIRE-IHDJ), as well as Mpox clade IIb (MPXV/2022/FR/CMIP) and Mpox clade Ib (MpxV/PHAS-506/Passage- 03/SWE/2024_09_11), were used for antibody neutralization assays. All assays were performed on U2OS cells, seeded at 2×10^4 cells per well in 96-well plates 24 hours prior to infection, using MV-enriched viral preparations as described above.”

11. Line 829. Do not understand if neutralization was done in the presence of 6% or 1.5% NHS? I believe the standard way of determining % complement would be the amount that is incubated with virus and antibody (prior to adding to cell culture to determine the remaining infectious titer).

We thank the Reviewer for this question. The neutralization assays in our study were performed by mixing virus with 6% NHS in the inoculum. Following the addition of antibodies, the effective concentration was 3% in the antibody–virus mixture, which was pre-incubated for 2 hours. After a 1:1 dilution onto cells, this resulted in a final concentration of 1.5% NHS in the infected wells. While both approaches to reporting complement concentration are commonly used in the literature (either (i) the concentration present in the inoculum prior to cell addition, or (ii) the final concentration in the infected wells), to enhance reproducibility, we described the complete workflow in the Methods, which may have caused confusion regarding the effective concentration. To avoid ambiguity, and to address the Reviewer’s comment we now clearly state in the figure legends that the NHS percentages reported correspond to the final concentration in the infected wells.

“The percentages of complement sources indicated represent the final concentrations in the infected wells.”

12. Line 843. Similarly, do not understand if mpox neutralization was done in the presence of 40% or 10% guinea pig complement?

Similarly, as explained in answer to comment 11, the Mpox neutralization assays were performed by mixing virus with 40% GPC in the inoculum. Following the addition of antibodies, the effective concentration was 20% in the antibody–virus mixture, which was pre-incubated for 2 hours. After a 1:1 dilution onto cells, this resulted in a final concentration of 10% GPC in the infected wells. To avoid ambiguity, we will clearly state in the figure legends that the GPC percentages reported correspond to the final concentration in the infected wells.

“The percentages of complement sources indicated represent the final concentrations in the infected wells.”

13. Line 908. What was the source of IHDJ virions were used (e.g., mainly MV or could this be EV)?

The VACV IHDJ virions used in this assay were MV-enriched preparations, obtained from viral stocks subjected to three freeze–thaw cycles. The relevant section was now updated in the Methods section (lines 945-948).

“MV-enriched purified VACV IHDJ virions ($\sim 5 \times 10^6$ PFU) were incubated with 62.5 $\mu\text{g/mL}$ of anti-Mpox mAbs and 1.5% NHS for 2 hours at 37°C. Following incubation, the virions were stained with anti-C1q-Alexa Fluor 647 (Santa Cruz) for 1 hour. PFA was then added to the virions to reach a final concentration of 4%, followed by analysis of the GFP-expressing virions in the Flow Cytometer.”

14. Figure 1B. Why such a large size difference between the expressed mpox and VACV A33 protein?

We thank the Reviewer for raising this point. VACV A33 appears larger than Mpox A35 on Western blot, which can be explained by two non-exclusive factors: first, the expressed VACV A33 ectodomain (residues 58–186) is 5 amino acids longer than the corresponding Mpox A35 construct (58–181), modestly increasing its molecular weight; second, because both proteins were expressed in a mammalian system capable of N-linked glycosylation, differences in glycosylation may also contribute, as Mpox A35 contains one consensus N-glycosylation site (NKS at position 78), whereas VACV A33 contains two (NCT at position 68 and NKS at position 78), providing one additional site that would increase its apparent size.

15. Figure 1H, table. Why does first column show 8M2110 and 10M2146 not competing, but second column show they do compete? Later crystal structures show binding sites are close and authors state this explains the competitive binding observed

*We thank the reviewer for this comment. As the reviewer correctly pointed out, the anti-A28 mAbs 8M2110 and 10M2146 do compete; however, this competition depends on the orientation of the assay, as shown in Figure 1H. When 8M2110 binds A28 first, 10M2146 is able to bind second. In contrast, when the reverse orientation is tested and 10M2146 binds A28 first, 8M2110 is not able to bind. The competition between the mAbs can be explained by two factors: the first, and less significant, is a small overlap between their binding residues as seen in Figure 4E; the second, and more substantial, is a steric clash between the variable domain of the light chain of 8M2110 and the variable domain of the heavy chain of 10M2146 (This was now added as **new Panel F to Figure 4**). We assume that this competition is more prominent in the second orientation (when 10M2146 is bound first and 8M2110 attempts to bind), likely due to significantly stronger binding of 10M2146 to A28 compared to 8M2110 (Figures 1E-1G).*

The relevant section was now updated in the Results section (lines 296-301).

“Mapping the footprints of both antibodies on A26 (Figure 4E) reveals that they bind two distinct epitopes that are very close to one another, with some overlap on helix $\alpha 11$. Superimposition of the complexes demonstrates a steric clash between the variable domain of the 8M2110 light chain and the variable domain of the 10M2146 heavy chain (Figure 4F), thus explaining the inability of 8M2110 to bind 10M2146 pre-bound A28 in ELISA (Figure 1H).”

New Panel F to Figure 4. Superimposed A26–antibody complexes highlight a steric clash between 8M2110 and 10M2146. Superimposed structures of A26 bound to 8M2110 (HC in green, LC in light gray) and 10M2146 (HC in dark gray, LC in light gray). A26 is shown with its NTD in blue and CTD in tan. A steric clash occurs between the variable domain of the 8M2110 light chain and the variable domain of the 10M2146 heavy chain (boxed region), explaining the observed competition.

16. Figure 1 colored lines and colored data points on graphs. Shades of each color used is difficult to distinguish. Can the MAbs that moves forward be the darkest shade and perhaps their names bolded in the legend? Seems like anti-A26 MAbs 8M2110 and 10M2146 and anti-H3 7M1162 are the ones used in multiple assays.

We thank the Reviewer for this suggestion, and we completely agree. The color scheme of the anti-A28 was altered throughout the Figures to make them more distinguishable. Furthermore, the names of the neutralizing mAbs in Figure 2 were bolded for better visibility.

17. Figure 2C table. Are values in the table based on graph in Fig 2B or 2C or both? Reason I ask is anti-H3 MAb 5M1111 only looks to neutralize with 10% guinea pig complement (and not active with 1.5% human complement).

*We thank the Reviewer for this important comment. The IC_{50} data in the **original Figure 2** were derived from Figure 2B and 2C, where VACV neutralization assays were performed with NHS, while Mpox neutralization assays were performed with GPC. The reviewer was correct in pointing out that mAb 5M1111 only neutralized in the presence of 10% GPC (used in Mpox assays) but not with 1.5% NHS (used in VACV assays). Since mAb 5M1111 binds exclusively to the Mpox H3 protein and not to its VACV H3 homolog in ELISA, we suspected that the observed neutralization pattern reflected virus species specificity (Mpox vs. VACV) rather than the complement source (GPC vs. NHS). To address this point, we repeated the Mpox neutralization assays now using NHS. Under these conditions, mAb 5M1111 effectively neutralized Mpox, confirming that its neutralizing activity is species specific, rather than dependent on a specific complement source. This data is now included in the **revised manuscript Figure 2**.*

18. Figure 3. Legend for panel F is missing.

Thank you for noticing this. This issue was now corrected.

19. Figure 5F & 5G. Were there any controls of virions with complement but no antibody or a control non-binding antibody?

We thank the Reviewer for this question. C1q, as well as IgG and C3 deposition, were assessed in the presence of 1.5% NHS using anti-Mpox mAbs 7M1162 (anti-H3) and 8M2110 (anti-A28), alongside the isotype control antibody MGO53, under identical assay conditions. To complete the dataset, TEM images corresponding to the isotype control antibody have now been added to the revised Figure 5 as Figure 5H.

20. Supplemental figure 1. Title of figure should not include IHDJ.

This issue was now corrected in Supplemental figure legend 1.

21, Supplemental Figure 3. Might be nice to include sequence for Copenhagen homolog to help explain non-binding of MAbs described in this paper.

We thank the Reviewer for this suggestion, and we completely agree. The sequence alignment of both Mpox clade Ib and VACV Copenhagen strain have now been added to Supplemental figure 3 and its legend has been updated accordingly (lines: 540-546)

“A28 homologs (OPG153) from six representatives Orthopoxviruses—Mpox virus sub-clade IIb (MPXV; accession A0A7H0DNC4), Mpox virus sub-clade Ib (Ref-SKU: 012V-06039) Variola virus (VARV; Q89489), Vaccinia virus WR (VACV; P24758), Vaccinia virus Copenhagen (VACV; P21114.1), and Camelpox virus (CMPX; Q8QQ29). Secondary structure elements are labeled above the sequences and colored by domain (NTD in blue, CTD in tan). Strictly conserved residues are highlighted with a red background. Epitope residues (BSA > 10 Å²) are marked by color-coded spheres below the sequences: red for 10M2146 and yellow for 8M2110.”

Reviewer #3 (Remarks to the Author):

In this manuscript, the authors isolated antibodies to the Mpox antigens A28, A35, and H3 from convalescent donors. They determined that anti-A28 antibodies were able to neutralize Mpox and VACV through complement-dependent mechanisms and showed the crystal structures of two anti-A28 antibodies to the VACV homologue A26. The binding sites were distinct but highly conserved between VACV and Mpox. They also show that vaccination with Mpox A28 produced antigen-specific B-cells and high neutralization responses to VACV infection in mice. Overall, I found the biological data to be well done and convincing. The findings are also timely and valuable to the field. However, there are some issues with the structural data that need clarification and refinement prior to publication.

We thank the Reviewer for their time and effort to evaluate our study, and for the thoughtful assessment and positive comments, particularly regarding the quality and rigor of the biological data. As outlined below, we have carefully addressed all the points raised by the Reviewer.

Major issues:

1) Some reported statistics in Table S3 are below acceptable standards. For the A26 + Fab 10M2146 data set the authors report an of 0.5 in the highest resolution shell. This can be acceptable if the CC1/2 is sufficient but the reported value is 0.21 in the highest resolution shell is really pushing it to the extreme limit suggesting the resolution is lower than reported. There are also numerous discrepancies between the text, Table S3 and the validation reports. Some differences are fine as the values in the validation report and values from phenix.refine or MolProbity are calculated differently but should be within reason. In the text the resolution for A26 + Fab 10M2146 is reported to be 1.9A but stated as 1.80A elsewhere. The validation report for A26 + Fab 10M2146 reports a of 1.27 with 0% Ramachandran outliers but Table S3 has these values of 0.5 and 2.34%, respectively. If the is actually 1.27 with a CC1/2 above 0.3 this resolution would be acceptable.

We thank the reviewer for this comment. We have reprocessed the data of the A26/10M2146 complex and decided to cut the resolution to 1.9 Å (CC1/2 = 0.68, I/s>1.8). We have reviewed the rest of the statistics in the table and ensured that they are consistent with the statistics reported in the validation reports. We have also added to the table the resolution at which I/s>2 (2.9 Å and 1.93 Å for the complexes A26/8M2110 and A26/10M2146, respectively), so that readers can judge for themselves the resolution of the data.

As requested by the reviewer, we have also included the number of TLS groups used in the refinement (1 group/chain) and explain it in the Methods section (lines: 1043-104).

“Structural models were built and refined iteratively using *phenix.refine* (Phenix version 1.19.2-4158)⁸⁵ and *Coot*⁸⁶, applying isotropic B-factor and one TLS group per chain.”

2) Starting with the paragraph on line 118 and Fig1 K-M discuss the role of antibody maturation and effects of germline revision on binding to Mpox or VACV. It would be useful to extend this analysis to the structures of 8M2110 and 10M2146. Do the somatically mutated residues explain why 8M2110 loses binding to Mpox A28 but not VACV A26 and why 10M2146 binding is unaffected? The inclusion of a sequence alignment of the mature and germline mAbs highlighting interacting residues would be helpful.

We thank the reviewer for this suggestion. To address the comment, we have included a new Figure S4 that indicates somatic mutations and paratope residues of both antibodies. Structural analysis does not provide a straightforward explanation for why the germline of 8M2110 binds to A26 and not A28. However, we think that the Serine in position 50 (that appears in the germline of 8M2110) alters the conformation of the loop in such a way that it clashes with V207 in A28 but not with A207 in A26. On the other hand, the somatic mutation changing Serine in that position to Threonine enables the formation of a new hydrogen bond between T50 of 8M2110 and H210 in A28, thus resulting in binding to A28 as well as A26. In 10M2146 there are three somatic mutations in the paratope, but none of them generate apparent

clashes or electrostatic repulsion, which explains why there are no binding differences between the germline and the mature antibody.

We have included this data in the Results section (lines 287-288 and 294-296).

“The same can be said of the three somatic mutations of the paratope, K78R, N121Y and A125S (Figure S4), which do not generate apparent clashes or electrostatic repulsions.”

“Indeed, the 8M2110 germline has a mutation close to CDR-H1 that could induce a conformational change incompatible with the presence of V207, which would explain why the germline does not bind A28 (Figure S4).”

3) With the major focus of the manuscript being on Mpox, can the authors comment on why the structure was not determined with Mpox A28. Was it tried and didn't work or was VACV A26 chosen initially and why was that used?

The main reason is that, in France, DNA fragments encoding for mpox proteins are subject to a special regulation and, to produce mpox A28, we need to apply for an authorization and follow a strict protocol for importing plasmids. This process would have delayed the structural experiments by several months. Given the high sequence similarity between A26 and A28, we have decided to conduct the structural studies with the VACV protein.

4) The inclusion of the epitope prediction and modelling of H3 and 7M1162 is confusing, not necessarily informative and has several issues. In the text the clusters are referred to as “clusters (1-3)” but in the figure they are clusters A-C. The methods say that the AlphaFold3 model was generated with Mpox H3 and 7M1162 but the VACV H3 model is shown in FigS4. Was AF3 used to predict the binding to VACV H3 and, if so, how does it compare to that of Mpox H3. How AF3 was used needs to be much more specific. Was the webserver used or local version? If a local version was used, how many models were generated and how many seeds were used? How was the binding model shown in FigS4 chosen, like what metrics were used to decide on this one? Was the model done with full Fab or just variable region of 7M1162?

We thank the Reviewer for this extremely important and relevant comment. In our initial submission, the AlphaFold3 prediction was performed using the webserver version with a single seed, and we presented only one model. This analysis was intended as a complementary and exploratory approach rather than the basis of our conclusions. To address the Reviewer's concern, we repeated the AlphaFold3 analyses using multiple seeds and all five generated models. These, however, produced variable and inconsistent results, with predicted epitopes located on different regions of the antigen. Nevertheless, our identification of histidine 167 as part of the 7M1162 binding site was guided by peptide mapping from phage display and subsequently confirmed by point-directed mutagenesis, and not by AlphaFold3 modeling. To improve clarity and avoid overinterpretation, we have now removed all AlphaFold3 analyses from the manuscript. The revised text focuses exclusively on the phage display-based prediction and mutagenesis validation, which provide a consistent and reliable basis for defining the epitope.

The relevant section was now removed from the Methods and supplemental figure legends sections updated in the Results section (lines 302-307) and.

“The binding epitope of the anti-H3 mAb 7M1162 was predicted using 11 affinity selected peptides isolated from screening random phage display peptide libraries followed by computational prediction⁵²⁻⁵⁵. This approach generated two predicted clusters (A and B) (Figures S6A-S6C). Point mutation in position 167 within Cluster A H3_{H167G}, H3_{H167K} exhibited reduced to complete loss of binding to 7M1162, thus supporting the critical role of this residue in the antibody binding site (Figure S6D). Notably, the H167K or H167G substitution did not affect binding of 5M1111.”

Minor issues:

1) *Throughout the figures, the slight differences in green color for the A28 binding antibodies is very hard to distinguish between.*

We agree with the Reviewer, this point was also raised by Reviewer 2. The color scheme of the anti-A28 was altered throughout the Figures to make them more distinguishable.

2) *The significant figures in Table S3 should be consistent. For example, the unit cell dimensions for one data set are reported to 1 decimal place but 3 in the other.*

We have checked and confirmed that all the figures of the updated Table S3 are consistent.

3) *It is reported that TLS refinement was used. How many TLS groups were used per chain?*

Thanks for the comment. We have used one TLS group per chain, 16 in the complex with 8M2110 and 4 in the complex with 10M2146.

We now indicate this in the Methods section (lines: 1033-1035) and in the updated Table S3.

4) *The methods mention the use of an anti-Fab VHH in the complex but none of the figures in the main text or supplemental show this anti-Fab VHH in the structural model. While not necessarily relevant to the conclusions of the paper, a figure showing the entire complex or contents of the ASU should be included.*

To address this comment, we have included a new Figure S5 showing the structure of the A26:Fab:VHH complexes.

5) *In the methods it says the initial phases were determined with a published the A26 model. Were the Fab or anti-Fab VHH included in phasing? If not, how were the Fab and anti-Fab VHH models initially built and placed?*

We thank the reviewer for the comment. As initial model for the anti-Fab VHH, we used the structure available in the PDB (7PIJ), and for Fabs we used models obtained with Alphafold.

We have now included this information in the Methods section (lines 1039-1043)

“Diffraction data were processed using XDS (version January 10, 2022)⁸³, and initial phases were obtained using PHASER software⁸⁴ with the A26 model (PDB: 6A9S)³⁵ the VHH model (PDB: 7PIJ, chain E) and a model of the Fabs obtained using Alphafold3 as search templates. In both complexes, we observed clear electron density for all the components, and we built models comprising A26, the Fabs and the anti-Fab VHH (Figure S5).”

6) *This manuscript was likely submitted before the PDB announcement of transitioning to Extended PDB IDs (<https://www.rcsb.org/news/feature/6875133f3b59581b68019794>). I would recommend updating all references to PDB ID to this extended format.*

We have now included the extended PDB IDs in the updated Table S3.

We appreciate the Reviewers' thoughtful and constructive re-evaluation of our revised manuscript, "**Potent Neutralization by Antibodies Targeting the Mpox A28 Protein**," submitted to *Nature Communications*. It is gratifying that the revisions and clarifications were found satisfactory and that the issues raised in the initial review have been fully resolved. The Reviewers' continued recognition of the study's novelty, therapeutic relevance, rigor, and high data quality is deeply appreciated.

Color code:

Reviewer's text

Authors' response

Changes in the text

Reviewer #1 (Remarks to the Author):

The authors have adquetaly addressed the Reviewers comments. The paper remains strong and of broad interest to both MPXV vaccinologists and the broader virology community.

We thank Reviewer #1 for their positive and encouraging feedback. It is gratifying that the revisions were considered satisfactory and that the study is recognized as both strong and of broad interest.

Reviewer #2 (Remarks to the Author):

Authors provide an excellent and detailed response to Reviewers' comments and have appropriately adjusted text in the manuscript.

We thank Reviewer #2 for their positive evaluation and for acknowledging the thorough and detailed responses to the Reviewers' comments. We are pleased that the revisions and manuscript adjustments were considered appropriate.

I appreciated the authors' explanation of how they report the concentration of complement used in their neutralization assays. However the reference they cite in the methods (ref 46; line 875/876) uses the complement concentration during incubation with virus prior to adding to cells to measure the amount of MV neutralized. I believe that would be the more standard way to report the % complement included during a virion neutralization assay. Note that there is approximately 2-log decrease in MV titer within the first 30 mins of incubation of vaccinia virus with antibody and complement (PMID 1731333). Also note that ref 75 may not be a correct reference in the methods section.

We thank the Reviewer for their careful reading and insightful comments. References 46 and 75 were cited to describe the methodology ultimately used in this study, as developed by Hubert et al. (ref. 46) and further expanded in Postal et al., (ref. 75) through the use of additional Mpox sub-clades, including sub-clade Ib used in this manuscript. The complement source and percentage used in our assays differs from those references, and this has been clearly indicated in the Methods for transparency.

Regarding the Reviewer's comment about our presentation and reporting of the complement percentages in the manuscript: the concentrations reported for complement, virus and antibodies are the final concentrations inside the well, while, as the Reviewer noted, during the pre-incubation the concentration of all three is double. This has been clearly described in Methods. In the Main Text we report the complement percentage of our assays as the "final concentration in the infected wells" to avoid any confusion and misinterpretation. While in Figure 5 we show that in the case of our mAbs most of the neutralization occurs during pre-incubation period, we feel that indicating the final concentration in the infected wells is more appropriate for the sake of consistency. Please note, that antibody concentration is conventionally reported based on the final concentration inside the well. As to satisfy the Reviewer's request, we added to all figure legends throughout the manuscript the complement concentration during pre-incubation as well as the final concentration in the well (For example in lines 1220-1222):

"Anti-MPXV mAbs (excluding anti-A28 mAbs 8M2110 and 10M1135) neutralization of MPXV sub-clade IIb in the absence and presence of 10% GPC (20% during pre-incubation)."

Minor comments to consider to enhance current text and focus mostly on the abstract.

1. Line 28 (and 68) Curious about describing A26 as a virulence factor when deletion of the gene actually increased virulence when compared to the parental virus in mouse model. Nuanced description of A26 functions in the discussion is more accurate. Authors should consider A28 as a key or important virion protein.

We thank the Reviewer for this suggestion and agree. Therefore, the formatting in the Abstract (lines: 27-28) and Introduction have been changed accordingly (lines 69-71):

"We isolate and characterize a panel of monoclonal antibodies (mAbs) targeting MPXV A28 (OPG153), an important membranal protein present on mature MPXV virions."

"Acting as a key membranal protein, A28 facilitates viral attachment through laminin binding and controls endosomal entry by regulating the viral fusion machinery^{19,35,36"}

2. Line 33. Abstract says, MAb attenuated disease in infected mice, but then later (line 80 or section starting on line 391), the MAb treated mice yielded only modest, non-significant clinical benefit in vivo. Perhaps better said, Passive transfer of 8M2110 had a trend toward attenuating disease in infected mice.

We appreciate the Reviewer's comment and agree with the recommendation. The Abstract has been revised accordingly to reflect this clarification (lines 33-34):

"Passive transfer of 8M2110 modestly attenuates disease in infected female mice."

3. Line 35 (and 408) Would not characterize the protection provided by active immunization as complete given significant weight loss. Would just say protection.

We thank the Reviewer for this comment and agree. The description has been revised to indicate “protection” rather than “complete protection” in the Abstract (lines 34-35) and the Results section (282-283):

“Moreover, immunization with A28 elicits antigen-specific B cells and robust neutralizing antibody responses and provides protection against lethal VACV challenge.”

“Immunization with MPXV A28 conferred protection against a lethal VACV challenge (Fig. 6b-c and Supplemental Fig. 10a).”

4. Line 58/59. Would include OPG names for A29, E8, H3, M1, A35 and B6

We thank the Reviewer for this suggestion. The OPG names for A29, E8, H3, M1, A35 and B6 have now been included in the Introduction (lines 58-61):

“These include the MV attachment and fusion proteins A29 (OPG 154), E8 (OPG 120), H3 (OPG 108), M1 (OPG 95), A16 (OPG 143), G9 (OPG 94) and the EV proteins A35 (OPG 161) and B6 (OPG 190—MPXV nomenclature)^{22,24-30}.”

5. Line 198/199. Based on new Figure S3, looks like Copenhagen A26 had a frameshift mutation that results in altered C-terminus amino acid sequence as well as a truncation.

We thank the Reviewer for this valuable observation. Indeed, our analysis indicates that VACV Copenhagen A26 carries not only a C-terminal truncation but also multiple deletions and frameshift mutations that alter the amino acid sequence preceding the truncation (residues 196–322). Interestingly, this region appears to overlap with coding sequences of other putative proteins in VACV WR, suggesting that the deletion in Copenhagen A26 may result in a fusion with additional viral proteins. Nevertheless, the loss of the C-terminal domain prevents A26 from interacting with A17, thereby excluding it from the virion surface. Consistent with this, even mAb 10M2146—whose epitope lies almost entirely within the conserved region of A26—fails to neutralize VACV Copenhagen. While this finding is intriguing, it lies beyond the scope of the current study.

6. Line 688. What experiments used sucrose purified MV particles? Was it just virions used for electron microscopy. So instead of saying experiments, would say here what experiments used lysate that were pelleted through a sucrose cushion.

We appreciate the Reviewer’s comment and agree with the recommendation. The Method section has been revised accordingly to reflect this clarification (lines 420–423):

“For the immunogold labeling and transmission electron microscopy experiments, purified MV particles, infected cell lysates were centrifuged and overlaid onto a 36% sucrose cushion, followed by ultracentrifugation at 33,000 × g for 80 minutes. The resulting pellet was then resuspended in 10 mM Tris-HCl (pH 9).”

7. Authors' response letter have a few revised Figure panels that has panels that are not in the manuscript.

We thank the Reviewer for this observation. Indeed, the response letter includes additional experimental data generated to address specific Reviewer comments, some of which were not included in the revised manuscript. Some of these findings extend beyond the scope of the present work and will be further investigated in a follow-up study focused on detailed evaluation of A28 as an immunogen.

Reviewer #3 (Remarks to the Author):

The authors have adequately addressed all of my comments and concerns. This is a very well done study.

We thank Reviewer #3 for their positive assessment and thoughtful evaluation of our work. We appreciate the recognition of the study's quality and the careful consideration of our responses.